# REPLAY CAN PROVABLY INCREASE FORGETTING

## ABSTRACT

Continual learning seeks to enable machine learning systems to solve an increasing corpus of tasks sequentially. A critical challenge for continual learning is forgetting, where the performance on previously learned tasks decreases as new tasks are introduced. One of the commonly used techniques to mitigate forgetting, sample replay, has been shown empirically to reduce forgetting by retaining some examples from old tasks and including them in new training episodes. In this work, we provide a theoretical analysis of sample replay in an over-parameterized continual linear regression setting, where given enough replay samples, one would be able to eliminate forgetting. Our analysis focuses on replaying a few examples and highlights the role of the replay samples and task subspaces. Surprisingly, we find that forgetting can be non-monotonic with respect to the number of replay samples. We construct tasks where replay of a single example can increase forgetting and even distributions where replay of a randomly selected sample increases forgetting on average. We provide empirical evidence that this is a property of the tasks rather than the model used to train on them, by showing a similar behavior for a neural net equipped with SGD. Through experiments on a commonly used benchmark, we provide additional evidence that performance of the replay heavily depends on the choice of replay samples and the relationship between tasks.

## 1 INTRODUCTION

Humans and other animals can seemingly learn new skills and accumulate knowledge throughout their lifetimes. Continual learning algorithms aim to achieve this same capability: to produce systems that can learn from a sequence of tasks. One of the main challenges is a phenomenon typically called catastrophic forgetting (McCloskey & Cohen, 1989), where the learner's performance on a previously visited task degrades once it learns new tasks. A dominant theme in continual learning has been the development of methods to address catastrophic forgetting (Li & Hoiem, 2017; Zenke et al., 2017; Rebuffi et al., 2017; Kirkpatrick et al., 2017; Prabhu et al., 2020b). However, there has been limited theoretical treatment of their efficacy.

In this work, we focus on a continual learning problem consisting of a sequence of linear regression tasks. The tasks are designed such that a single linear model is sufficient to solve the full sequence. The aim of focusing on this problem is that it is simple enough to permit analysis while preserving some key challenges of the continual learning problem. A particularly noteworthy discovery of prior work is that even in this setting, catastrophic forgetting can occur (Evron et al., 2022). However, an open question remains: Can methods designed to combat forgetting succeed in this setting? As a first step towards answering this question, we study one such method, experience replay.

There are many variations of experience replay. For example, van de Ven et al. (2020) introduce a brain-inspired generative replay and show that it has strong performance on complex benchmarks. In this paper, we focus on sample replay, an intuitive technique where samples observed during continual learning are stored and repeated back to the learner during later tasks, to help retain solutions to prior tasks. This simple method has been shown to be effective at ameliorating forgetting (Rebuffi et al., 2017; Rolnick et al., 2019; Aljundi et al., 2019b; Wu et al., 2019; Chaudhry et al., 2019; Tiwari et al., 2022), and is often used in dynamic learning settings like reinforcement learning (Lin, 1992). Replay has also been a focus of study in neuroscience, as strong experimental evidence supports the hypothesis that replay plays an important role in memory consolidation (Rasch & Born, 2007; Oudiette & Paller, 2013).

Some existing theoretical works in continual learning show that when tasks are revisited in a cyclical order, forgetting would vanish. Evron et al. (2022) shows this in the continual linear regression setting, while Chen et al. (2022) show a similar result in a more general PAC-like continual learning setting. While these results show that forgetting decreases when the entire task sequence is replayed, they do not consider what happens when replay occurs between tasks, and involves a subset of samples. We analyze sample replay for continual linear regression and find that, when the number of replay samples is small, the outcome of sample replay can vary significantly. A learner faced with a possibly infinite sequence of tasks and finite memory cannot afford to hold on to a fixed number of samples per task, so it is important to understand this low sample storage setting. Even when storage capacity is not a concern, studying this low sample regime could lead to insights that make learning more efficient.

We first prove that somewhat counter-intuitively there are worst-case scenarios where replay actually causes more forgetting. We then prove a surprising stronger result, that this can still hold in a certain average case sense: even when examples are sampled randomly from specific task subspaces, and replay samples are chosen randomly, replay can increase forgetting on average. These findings suggest that when the memory allocated to storing replay samples is very limited, not only does the choice of replay samples matters, but also the relationship between tasks could dictate the effect of replay on forgetting. In addition to our theoretical contributions, we provide an empirical investigation of forgetting with sample replay to support our theoretical findings. We verify our theoretical results and further show that the same surprising behavior exists in continual linear regression learning with neural networks. Through experiments on MNIST continual learning benchmarks, we show that there is significant variation in the effectiveness of replaying a few samples to mitigate forgetting, and a task sequence where replay can increase forgetting.

## 2 BACKGROUND AND SETUP

### 2.1 BACKGROUND

Evron et al. (2022) initiated the study of catastrophic forgetting in overparameterized linear regression. They consider a sequence of linear tasks $(\boldsymbol{X}_t, \boldsymbol{y}_t)_{t=1}^T$ where $\boldsymbol{X}_t \in \mathbb{R}^{n_t \times d}$, $\boldsymbol{y}_t \in \mathbb{R}^{n_t}$ and $n_t, d$ are the number of samples per task and input dimension respectively. They assume that the sequence of linear tasks share a solution that could be obtained by jointly training on all tasks, and for each task $n_t < d$, so any single task would not necessarily contain all the information needed to learn the common solution. Despite the existence of a common solution, they show that there are sequences of tasks such that learning them in a sequential manner with gradient descent will result in a significant amount of forgetting, which is defined to be the average error on all previously seen tasks (Definition 2.3). In this setting, $d$ many samples would be sufficient to recover the solution to each task.

Another group of techniques used to mitigate forgetting is through regularization. For example, forgetting can be eliminated using a Fisher information based weighting matrix (Kirkpatrick et al., 2017; Evron et al., 2023). We note that storing such matrices would take order $d^2$ bits of memory, which is of the same order as storing $d$ samples. Peng et al. (2023) introduce a general notion of an "ideal continual learner" and instantiate it for continual linear regression. Their algorithm maintains the shared null space of the previous tasks. Again, storing a null space could take order $d^2$ bits of memory. Sample replay, on the other hand, allows the learner to store much less than $d$ many samples, reducing memory requirements significantly. However, the effectiveness of replaying a few samples in this linear setting is an open question and the focus of this paper.

### 2.2 GENERAL SETUP

Here we consider two different settings for sample replay: the worst case and the average case. We first introduce the general setup, which is shared between the two settings and closely resembles the earlier formulation described above, and then examine each separately, in Sections 3.1 and 3.2.1, respectively. We start with the following two assumptions.

**Assumption 2.1** (Over-parameterized linear regression). We assume that each task is:

- Linear: there is $\boldsymbol{w}_t^* \in \mathbb{R}^d$ such that $\boldsymbol{X}_t \boldsymbol{w}_t^* = \boldsymbol{y}_t$.

- Over-parameterized: $k_t := \mathrm{rank}(\boldsymbol{X}_t) < d$.

We also assume realizability, which ensures that the $T$ tasks share a common solution:

**Assumption 2.2.** (Realizability). There exists $\boldsymbol{w}^* \in \bigcup_t \mathrm{span}(\boldsymbol{X}_t)$, where $\|\boldsymbol{w}^*\|_2 \leq 1$, such that for all tasks $t$, $\boldsymbol{y}_t = \boldsymbol{X}_t \boldsymbol{w}^*$.

**Projections.** Let $\boldsymbol{\Pi}_t$ be an orthogonal projection onto the row span of $\boldsymbol{X}_t$, i.e. the span of the samples of task $t$. We can write $\boldsymbol{\Pi}_i = \boldsymbol{X}_t^+ \boldsymbol{X}_t$ where $\boldsymbol{X}_t^+$ is the Moore-Penrose inverse of $\boldsymbol{X}_t^+$. Another way to obtain the orthonormal projection is using a matrix $\boldsymbol{W}_t$ whose columns form an orthonormal basis for the row span of $\boldsymbol{X}_t$. Given $\boldsymbol{W}_t$, we could write $\boldsymbol{\Pi}_t = \boldsymbol{W}_t \boldsymbol{W}_t^\top$. We use $\boldsymbol{P}_t := \boldsymbol{I} - \boldsymbol{\Pi}_i$ to denote the orthogonal projection onto the null space of task $t$. We use the term projection interchangeably with orthogonal projection throughout the rest of this paper.

**Learning Procedure.** Initially $\boldsymbol{w}_0$ is set to the all-zero vector. For each task $t$, starting with the solution $\boldsymbol{w}_{t-1}$ from the previous task(s), the learning algorithm minimizes squared error $\|\boldsymbol{X}_t \boldsymbol{w} - \boldsymbol{y}_t\|_2^2$ using GD or SGD.

It is known that training with GD or SGD leads to a solution that has minimum distance to initialization (Gunasekar et al., 2018; Zhang et al., 2021), that is,

$$\boldsymbol{w}_t = \arg\min_{\boldsymbol{w}} \|\boldsymbol{w} - \boldsymbol{w}_{t-1}\|_2 \ \text{ s.t. } \boldsymbol{X}_t \boldsymbol{w} = \boldsymbol{y}_t. \tag{1}$$

Parameter error of the procedure in Equation 1 satisfies the following recursive relationship

$$\boldsymbol{w}_t - \boldsymbol{w}^* = \boldsymbol{P}_t(\boldsymbol{w}_{t-1} - \boldsymbol{w}^*). \tag{2}$$

We include a derivation of this relationship in Appendix A.1 for completeness. Initially, the parameter error vector is $\boldsymbol{w}_0 - \boldsymbol{w}^* = -\boldsymbol{w}^*$, Equation 2 states that after training on task $t$, the parameter error vector is projected onto the null space of task $t$. So the parameter error vector evolves as a sequence of orthonormal projections into task null spaces, while forgetting also takes into account projection of the parameter error onto the task samples.

**Definition 2.3** (Forgetting). Given a sequence of training samples for tasks $S = ((\boldsymbol{X}_t, \boldsymbol{y}_t))_{t=1}^T$, the forgetting with respect to the training samples is defined to be

$$F_S(\boldsymbol{w}_T) = \frac{1}{T-1} \sum_{t=1}^{T-1} \|\boldsymbol{X}_t \boldsymbol{w}_T - \boldsymbol{y}_t\|_2^2. \tag{3}$$

We drop the subscript $S$ when it is clear from the context which sequence of tasks the forgetting is being computed over. Note that the average forgetting defined above is over $T-1$ tasks since forgetting on the last task is always zero. We can consider forgetting for a certain task ordering catastrophic, when $\lim_{T \to \infty} F_S(\boldsymbol{w}_T) > 0$, or in other words, when it does not vanish with the number of tasks.

*Remark* 2.4. In our average case result, each task is given by a distribution, and forgetting is measured on new samples from previous tasks' distributions. We introduce and discuss these details in Section 3.2.1.

Forgetting for the output of the learning procedure described in Equation 1 can be written as

$$F_S(\boldsymbol{w}_T) = \frac{1}{T-1} \sum_{t=1}^{T-1} \|\boldsymbol{X}_t(\boldsymbol{w}_T - \boldsymbol{w}^*)\|_2^2 = \frac{1}{T-1} \sum_{t=1}^{T-1} \|\boldsymbol{X}_t \boldsymbol{P}_T \boldsymbol{P}_{T-1} \dots \boldsymbol{P}_1 \boldsymbol{w}^*\|_2^2, \tag{4}$$

see Appendix A.2 for this derivation. We can see from Equation 4 that forgetting not only depends on the parameter error vector but also on its relationship with the training samples.

**Replay.** We consider a simple and standard formulation of replay in the literature, where the learning algorithm can store up to $m$ samples from the previously seen tasks in memory, sometimes called episodic memory (Chaudhry et al., 2019). Let $\boldsymbol{X}^{\mathrm{mem}}, \boldsymbol{y}^{\mathrm{mem}}$ denote the set of stored samples. During training on the current task $t$, in addition to the current task's samples, the model also trains on $\boldsymbol{X}^{\mathrm{mem}}, \boldsymbol{y}^{\mathrm{mem}}$ to get the new iterate $\tilde{w}_{t+1}$. There are many ways the algorithm can update $\boldsymbol{X}^{\mathrm{mem}}, \boldsymbol{y}^{\mathrm{mem}}$ (Chaudhry et al., 2019). In our worst case setup, this choice is adversarial, while in the average case setup we consider a random selection.

## 3 REPLAY CAN PROVABLY INCREASE FORGETTING IN CONTINUAL LINEAR REGRESSION

In this section we show that replay can increase forgetting in two different settings. Each setting demonstrates a different scenario where replay can increase forgetting. The worst case setting highlights how the relationships between individual samples could lead to catastrophic forgetting with replay, while the average case result goes beyond interactions between individual samples and highlights the role of task subspaces and the angle(s) between them. For both of these results, interference within samples of each task plays an important role. Since samples within a task can be revisited many times during training, this intra-task sample interference does not matter without replay and we only see forgetting due to interference across tasks. With replay, however, intra-task sample interference could also contribute to forgetting.

### 3.1 WORST CASE: FROM VANISHING TO CATASTROPHIC FORGETTING VIA REPLAY SAMPLE SELECTION

Our result in the worst case setting shows that the increase in forgetting due to replay can be rather dramatic, or catastrophic. In addition to Assumptions 2.1 and 2.2, we require that samples have unit norm:

**Assumption 3.1.** Let $X_{ti}$ be the $i$th row of $X_t$. Assume that $\|X_{ti}\|_2 = 1$.

By restricting the sample norms in Assumption 3.1, we are emphasizing that sample norms are not playing a role in the construction used to get the result in Theorem 3.2. This assumption is not necessary to get the worst case result. In the worst case setting, the adversary is free to choose the samples for each tasks, $((X_t, y_t))_{t=1,...T}$ and the replay sample(s) $X^{\text{mem}}, y^{\text{mem}}$, with restriction that $((X_t, y_t))_{t=1,...T}$ must satisfy Assumptions 2.1, 2.2, and 3.1.

#### 3.1.1 WORST CASE RESULTS

In this worst case setting, we choose the tasks and a (possibly empty) subset of samples from each task to be replayed. In this setup, we show that forgetting could increase from vanishing, $\mathcal{O}\left(\frac{1}{T}\right)$, to $\Theta(1)$, which we have labeled catastrophic.

At the core of the tasks constructed in the worst case setting are three samples $x_1, x_2$ and $x_3$, where $x_1$ and $x_3$ are orthogonal to one another, and $x_2$ is linearly independent but not orthogonal to $x_1$ or $x_3$. Consider a sequence of two tasks where the samples for the first task consists of $x_1, x_2$, and the second task contains only $x_3$. After training on the first task, there is no error on $x_1$ and $x_2$. When we train on the second task, since $x_3$ is not orthogonal to $x_2$, training on $x_3$ introduces some error on $x_2$ but doesn't introduce error on $x_1$, since it is othogonal to $x_3$. Now suppose that $x_2$ is replayed, so for the second task, we train on both $x_2$ and $x_3$. Since $x_1$ is not orthogonal to $x_2$, this will introduce error on $x_1$. Thus in this three sample, two task setup, replay causes an *exchange* of the error on $x_2$ with error on $x_1$. See Figure 7 in Appendix B.1 for a geometric illustration of this phenomena.

**Theorem 3.2** (Worst case replay). *Under assumptions 2.1 and 2.2, for any $T \geq 2, d \geq 3$, there is a sequence of $T$ tasks and a sample $(\tilde{x}, \tilde{y})$ such that without replay, forgetting is $F(w_T) = \mathcal{O}\left(\frac{1}{T}\right)$, while with replay of $(\tilde{x}, \tilde{y})$, forgetting is catastrophic, i.e., $F(w_T) = \Theta(1)$.*

In the proof, which is given in Appendix C.1 , we construct a scenario where this type of error exchange is detrimental. We construct a sequence of tasks $t = 1, \ldots, T$, where the sample $x_1$ occurs in all but the last task, while the sample $x_2$ occurs in just one of these tasks. Then we can see that if $x_2$ is replayed in the last task ($t = T$), causing error on $x_1$, the forgetting will be much larger, growing with the number of tasks $T$, since $x_1$ occurs in almost all the tasks. Note that in this construction, replaying $x_1$ would not change the final iterate $w_T$, since without replay, there is no error on $x_1$. Additionally, the construction is such that without replay, there is no forgetting on any sample other than $x_2$.

### 3.2 AVERAGE CASE: FORGETTING IN A RANDOM SAMPLE SETTING

The goal of studying sample replay in the average case setting is to understand how much the increase in forgetting due to sample replay depends on the relationship of individual samples within each

subspace and the choice of samples to be replayed. As we will see, the relationship between the tasks also affects forgetting with replay. We show that it is possible to pick task subspaces such that replay increases forgetting, even when task samples are chosen randomly from those subspaces and a replay sample is chosen randomly.

### 3.2.1 AVERAGE CASE SETUP

The average case setup is more natural and general relative to the worst case setup. Each task's samples are drawn from a distribution supported on some subspace, and the forgetting is measured on a set of new samples from that distribution. Additionally, the replay samples are chosen randomly.

In the average case construction, the task distributions are Gaussians supported on specific subspaces. Each of these subspaces can be specified by an orthonormal basis $W_t \in \mathbb{R}^{d \times k_t}$. The rows of the $n_t \times d$ dimensional matrix $\mathbf{X}_t$ consist of individual samples $\mathbf{X}_{t1}, \ldots, \mathbf{X}_{tn_t}$, where each sample

$$\mathbf{X}_{tj} = W_t \mathbf{Z}_{tj} \ \text{ and } \ \mathbf{Z}_{tj} \sim \mathsf{N}(0, \frac{\mathbf{I}_{k_t}}{k_t}), \tag{5}$$

and $\mathbf{Z}_{tj}$ are iid for $j \in [n_t]$. Since $\mathbf{X}_{tj}$ are rows of $\mathbf{X}_t$, we can write $\mathbf{X}_t = \mathbf{Z}_t W_t^\top$, where $\mathbf{Z}_t$ is a $n_t \times k_t$ dimensional matrix whose rows are $\mathbf{Z}_{tj}$. Then $w^*$ along with $\mathbf{X}_t$ determines $y_t = \mathbf{X}_t w^*$.

In this average case setting, we need to have enough samples for each task to span each task subspace $W_t$. The sample generation process described above ensures that this condition is met as long as the number of samples for each task is larger than the task's rank.

**Assumption 3.3.** Assume that the number of the samples for each task is at least as large as the rank of the given subspace $W_t$, that is, $k_t \leq n_t$.

While it will always be the case that $\operatorname{rank}(\mathbf{X}_t) \leq n_t$, the condition above is to ensure that $\operatorname{rank}(\mathbf{X}_t) = k_t$. We consider the expectation of forgetting with respect to $k_t$ test samples from previous tasks. To mark this difference we use $\mathbf{X}'_t, \mathbf{y}'_t$ to denote the test samples and $F_{S'}(w_T)$ to denote forgetting with respect to these new samples. That is

$$F_{S'}(w_T) = \frac{1}{T-1} \sum_{t=1}^{T-1} \|\mathbf{X}'_t w_T - \mathbf{y}'_t\|_2^2 = \frac{1}{T-1} \sum_{t=1}^{T-1} \|\mathbf{X}'_t(w_T - w^*)\|_2^2. \tag{6}$$

We assume that there are $k_t$ test samples. This would ensure that there are enough test samples to span the task's subspace. Next proposition gives the expected forgetting (without replay) in this setting, the proof is given in Appendix C.2.

**Proposition 3.4.** *Suppose that* $\mathbf{X}_{tj}$ *are sampled according to Equation 5, then*

$$\mathbb{E}[F_{S'}(w_T)] = \frac{1}{T-1} \sum_{t=1}^{T-1} \|\mathbf{\Pi}_t \mathbf{P}_T \ldots \mathbf{P}_1 w^*\|_2^2. \tag{7}$$

*Remark* 3.5. Even though Evron et al. (2022) did not have any distributional assumptions on the samples, the expected forgetting in the expression above is similar to the upper bound they get on forgetting, which was $\frac{1}{T} \sum_{t=1}^{T} \|\mathbf{\Pi}_t \mathbf{P}_T \ldots \mathbf{P}_1 w^*\|_2^2$. When it comes to replay, however, this setting introduces new challenges, since the projections into the null spaces would then depend on the replay samples and could be random in this average case setting.

### 3.2.2 AVERAGE CASE RESULTS

We now present the main result in Theorem 3.6, which states that replay can increase forgetting even in this setting. In the proof of Theorem 3.6, we first give an analysis of expected forgetting with replay for two tasks and then a simple two task construction where we can show that replaying a single randomly chosen example from the first task's examples increases forgetting on average. The construction is such that the first task has rank two and hence replaying two samples would lead to zero forgetting.

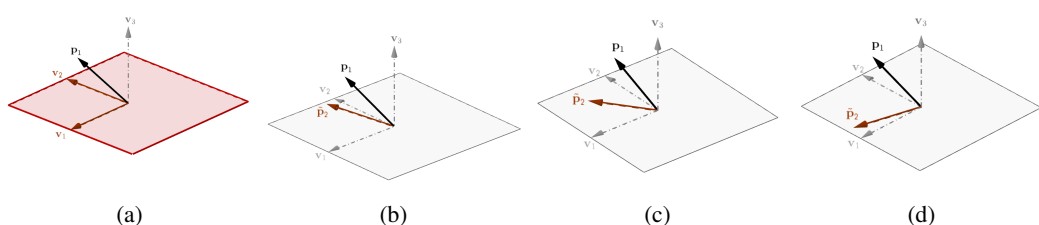

|     |     |     |     |
| --- | --- | --- | --- |
| (a) | (b) | (c) | (d) |

Figure 1: The red plane in (a) shows the null space of task 2. The null space of the first task, spanned by $p_1$, is in the span of $\{v_2, v_3\}$ and very close to $v_2$. Without replay, the angle between $v_2$ and $p_1$ determines forgetting , while with replay of one sample, the angle between $\tilde{p}_2$ and $p_1$ determines forgetting. In displays 1b and 1c where $\tilde{p}_2$ is not too far from $v_2$, forgetting would increase with replay, while in 1d, it would decrease.

**Theorem 3.6** (Average case replay). *Suppose that assumptions 2.1, 2.2, and 3.3 hold. For every $w^* \in \mathbb{R}^d$ where $\|w^*\|_2 \leq 1$, there exists a sequence of two task subspaces, such that replaying a randomly chosen sample from the first task's samples increases expected forgetting. That is*

$$\mathbb{E}[F_{S'}(w_2)] < \mathbb{E}[F_{S'}(\tilde{w}_2)], \tag{8}$$

*where $\tilde{w}_2$ is the iterate after the second task with replay.*

This result shows that there can be task sequences where most choices of sample options for replay are unfavorable, so that it is not only choices of replay samples that matters but also the relationship between the tasks.

*Remark* 3.7. A similar statement would hold if forgetting was measured with respect to the training samples. So this increase in forgetting is not due to overfitting. We have stated it with respect to the test samples, since it a stronger more general statement.

The proof of Theorem 3.6 is given in Appendix C.1. Note the statement in Theorem 3.6 is on the expectation of forgetting with replay, and it does not mean that replaying any sample from the first subspace would increase forgetting. In fact, there are directions in the first task's subspace such that replaying a sample in those directions would reduce forgetting.

Now we give an overview of our average case construction, and the intuition behind it. Fix an orthonormal basis $v_1, v_2, v_3$ of $\mathbb{R}^3$ and consider two tasks in $\mathbb{R}^3$, where the first and second tasks' null spaces has rank one and two respectively. Let the unit vector $p_1$ span the first task's null space. $p_1$ is chosen such that it is in the span of $\{v_2, v_3\}$, is very close to $v_2$. The second task's null space is spanned by $\{v_1, v_2\}$. See Figure 1a for an illustration of the null spaces. Replaying a sample in this setting would reduce the rank of the second task's null space. But this doesn't necessarily mean that the forgetting will be smaller. It is known that forgetting depends on the angle between the task null spaces in a non-monotonic way, see Appendix A.3 for more details. Initially, this angle is the angle between $p_1$ and $v_2$, which is very small. After replay, the second task's null space will be reduced to a one dimensional null space, spanned by $\tilde{p}_2$. Displays (b) and (c) in Figure 1 show some possible scenarios for $\tilde{p}_2$ where forgetting would increase, since the angle between $p_1$ and $\tilde{p}_2$ would be slightly larger than the angle between $p_1$ and $v_2$, but not much larger. Figure 1d on the other hand shows a scenario where forgetting would decrease with replay since the angle between $\tilde{p}_2$ and $p_1$ is much larger than the angle between $p_1$ and $v_1$. The construction in Theorem 3.6 is such that replay of most samples would result in cases like (b) and (c) in Figure 1.

*Remark* 3.8. Although it is not clear whether the conditions under which replay increases forgetting can be fully characterized, we note that a necessary but not sufficient condition is that there is some forgetting on the replay sample(s) to begin with. Otherwise, the gradient with respect to that sample would be zero and it would not have any effect on the iterate.

## 4 EXPERIMENTS

We have included three sets of experiments. The first set of experiments explores the extent to which our theoretical results hold empirically, in the more general settings of higher-dimensional linear

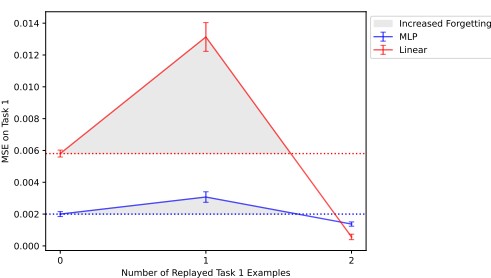 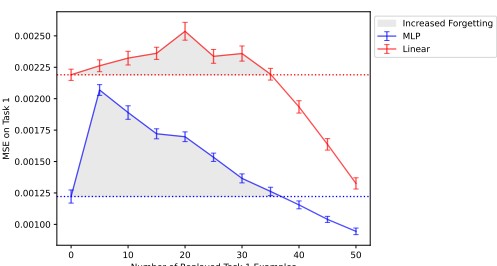

(a) The input data for the two tasks are given by the three dimensional construction given in Theorem 3.6. Each point is averaged over 150 runs.

(b) An extension of the three dimensional construction to $d = 50$ displaying a similar behavior. Each point is averaged over 60 runs.

Figure 2: Forgetting versus the number of replayed samples. Each plot shows forgetting of a linear model and a neural net with one hidden layer. We can see that forgetting initially increases with a small number of replay samples and then eventually decreases. The dashed lines show the baseline of no replay and the error bars indicate standard mean error.

regression, and with respect to more complex (non-linear) networks. The other two experiments involve MNIST (Lecun et al., 1998) and are in a classification setting. In the second set of experiments, we replay one sample and study how the class of the replayed sample affects forgetting. In the third set, we compare replay of different numbers of samples for two pairs of related task sequences.

## 4.1 EMPIRICAL EVALUATION AND EXTENSION OF THE THEORETICAL RESULTS

So far, we have shown that there are tasks that are realizable by linear models where sample replay increases forgetting when a linear model is trained sequentially. In this set of experiments, we verify these findings empirically and show that this behavior is not restricted to training with linear models. The experiments here show that there are (linear) tasks where replay can increase forgetting even when training nonlinear models on these tasks. We investigate the effect of replay on forgetting using a multi-layer perceptron (MLP) with one hidden layer and ReLU activations on a sequence of two tasks that are based on our worst case construction.

We consider two models: a linear model, and a MLP with one hidden layer. We consider two task sequence constructions. The first construction is in $\mathbb{R}^3$ and is based on the construction given in Theorem 3.6. The second task sequence is an extension of that construction into a higher dimensional space ($d = 50$). Each sequence of tasks consists of two tasks. The model is trained on the first task and then on the second task. Forgetting is then measured as the mean squared error of the final model on the first task's samples. When training with replay, $m$ samples from the first task are randomly selected without replacement and are combined with each batch during training on the second task. See Appendix D for further details on the experimental setup and how the construction in Theorem 3.6 was extended to a higher dimensional input.

Another experiment included in Appendix D.1.1, provides empirical evidence that forgetting for the nonlinear models in this setting is affected by the angle between task null spaces through a mechanism similar to linear models. This provides some insight into the behavior of the nonlinear models on the continual linear regression problem, as seen in Figure 2.

## 4.2 EXPERIMENTS ON MNIST

The goal of the experiments in this section is to translate the insights we have gained through the input task and data constructions in the linear setting to some simple but common benchmarks in continual learning. In all the experiments in this section, a MLP with two hidden layers of size 256 and ReLU activations was used. See Appendix D.2 for more detail on these experiments.

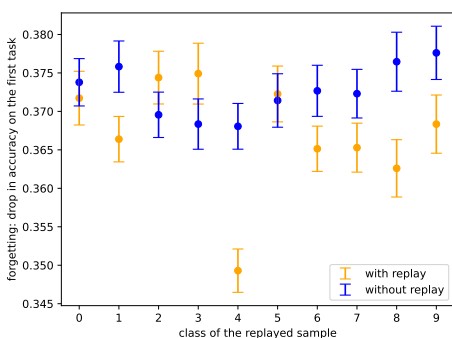 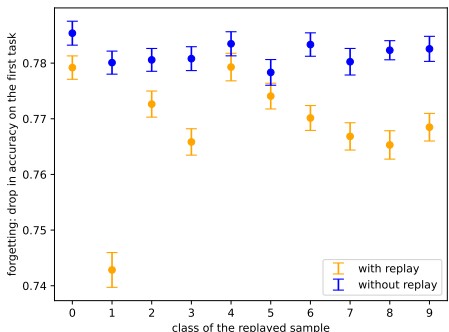

(a) 45 degrees rotation. Differences in means for classes 1, 4, and 8 are statistically significant.

(b) 90 degrees rotation. Differences in the sample means are statistically significant except for 4 and 5.

Figure 3: Class of the replayed sample affects forgetting. The x-axis shows the class of the replay sample while the y-axis shows the amount of forgetting in Rotated MNIST. The points depict the averages and the error bars show mean standard error over 80 runs. Comparing the average forgetting without replay to the one with replay of a single sample from each class shows that the effect of replay varies significantly across classes.

### 4.2.1 ROTATED MNIST

We study the role of replay samples in this experiment using Rotated MNIST (Lopez-Paz & Ranzato, 2017) in a task incremental setting. We consider two tasks, where the first task is MNIST and the second task has the same training data, except that the digits are rotated. Forgetting is measured as the drop in classification accuracy in test data from the first task. In each run, after training a MLP on the first task, a copy of the network is trained on the second task either without replay or with replay of a sample from the first task. The results of these experiments are shown in Figure 3. On average, the extent to which replay decreases forgetting depends significantly on the class of the replayed sample. For example, we can see that for 45 degrees rotation, replaying the digit 4 reduces forgetting by about 3%, while replaying the digit 5 doesn't seem to make much of a difference. Additionally, the relationship between tasks, in this case characterized by the degree of the rotation, also affects the behavior of replay with respect to the class of the replayed sample. For example, replaying a 4 seems to be more beneficial in the 45 degrees rotation case than the 90 degrees.

We take a closer look at replaying the digit 5 in the 45 degree rotation case, where replay seems to have minimal effect on forgetting. Figure 4 shows the distribution of differences in forgetting, that is, forgetting without replay minus forgetting with replay of a randomly selected digit 5 sample. We can see that even when on average this difference is not statistically significant, in many cases forgetting with replay exceeds without replay. These results suggest that there could be significant differences in the efficacy of replay depending on which examples are replayed.

### 4.2.2 SPLIT MNIST

In this experiment, we study whether the relationship between tasks affects forgetting with replay in a class incremental setting. In one task sequence the first task involves discriminating 0's from 1's, and the second 6's from 7's; in the other sequence the first task is 0 vs. 6, the second 1 vs. 7. Figure 5 shows that in the first task sequence replay consistently helps, but in the second forgetting initially increases, and only begins helping with 4 or more samples. We hypothesize that the visual similarity of the digits in the first task in the 0, 6 - 1, 7 sequence makes the forgetting worse with a small number of replay samples, as the challenging discrimination task is sensitive to the selection of samples.

This empirical result bears important similarities to our theoretical results, in the average case setting. The relationship between the tasks in a continual learning sequence determines the effect of replay on forgetting. And even in this more complicated, classification problem, there exists sequences where forgetting can increase with replay.

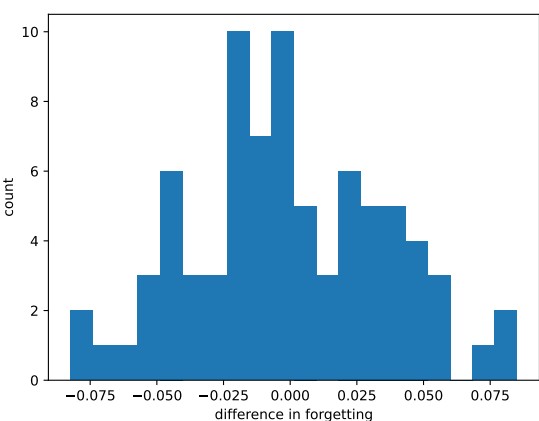

Figure 4: A histogram of differences in forgetting without replay and with replay of a digit 5 sample in Rotated MNIST, where the second task is rotated by 45 degrees.

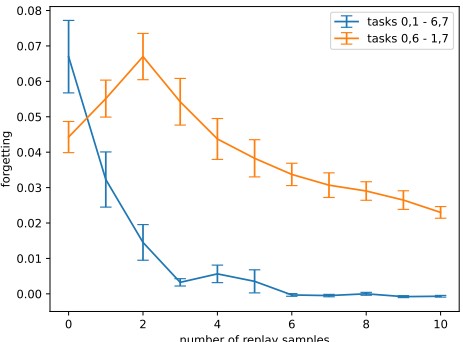

Figure 5: Role of tasks in replay for Split MNIST. For each task sequence, we have plotted the average forgetting against the number of replay samples. The error bars show mean standard error over 80 runs. Values for 0 replay samples show forgetting without replay. The differences in average forgetting across the two task sequences are statistically significant in all replay cases except for the no replay case. For the $0, 6 - 1, 7$ task sequence, the differences in the means for no replay and replay of 2 samples is statistically significant, so the observed increase in forgetting is not noise.

## 5 RELATED WORK

For general background on continual or lifelong learning, see the surveys by De Lange et al. (2022); Parisi et al. (2019); Wang et al. (2024). Theoretical studies of continual learning, and especially replay have been relatively more scarce and recent. Nevertheless, there are a few studies of continual learning in a linear setting. Doan et al. (2021) initiate the study of catastrophic forgetting of neural nets in the NTK regime, which also applies to linear models. Goldfarb & Hand (2023) study the effect of over-parameterization on forgetting in a linear regression setting for two tasks whose task subspaces are effectively low rank and picked randomly. Lin et al. (2023) study generalization and a slightly different notion of forgetting in continual linear regression. They allow the tasks to be realized by different linear functions, while the task subspaces are essentially random. Li et al. (2023) study the trade-off between stability and plasticity in a linear regression setting similar to (Evron et al., 2022), with the distinction that they use $\ell_2$- regularization while training on the second task. Shan et al. (2024) study continual learning in deep learning from a statistical-mechanics point of view.

Peng et al. (2023) define a general continual learner that has no memory constraints and incurs zero forgetting. They propose an instantiation of it for continual linear regression that could require up to order $d^2$ bits of memory. Additionally, through their framework, they derive uniform convergence type bounds and justify a form of sample replay where the learner keeps a task-balanced set of samples from previous tasks in memory and for each new task trains on the stored and new task's samples from scratch. Note that without knowing the exact constants, these bounds do not provide much information on replay of a few samples. Prabhu et al. (2020a) make a similar argument empirically and show that training from scratch on the samples stored in memory plus the most recent task's samples outperforms many methods that were specifically designed to address catastrophic forgetting.

There is a large body of empirical work on continual learning and methods used to mitigate forgetting, many of which use sample replay as a main strategy to address forgetting (Rebuffi et al., 2017; Chaudhry et al., 2019; Wu et al., 2019; Aljundi et al., 2019a; Caccia et al., 2021). Although there has been some concern in the literature that replaying a small number of samples might lead to overfitting (Lopez-Paz & Ranzato, 2017), Chaudhry et al. (2019) find that replay of even one sample per class improves performance and does not hurt generalization even when the model memorizes the replay samples. Verwimp et al. (2021) give a more complex picture of this in terms of the loss landscape. They find that while replay could keep the solution within a low loss region for previous tasks it could also pull the solution towards an unstable region. Motivated by forgetting in continual learning, Toneva et al. (2018) study forgetting of the examples across batches while training on a task. They find that there are complex examples that are prone to be forgotten across different model architectures. Prabhu et al. (2024); Zajac et al. (2023) show that their proposed replay-free methods achieve better performance than replay-based ones. Our results differ from these empirical results in that we focus on the existence of task sequences that would provably increase forgetting, and the variability of forgetting effects when the number of replay samples is small.

## 6 DISCUSSION

This work aims to initiate the theoretical study of sample replay in continual learning, complementing existing empirical work. We have shown that replaying a small number of examples can increase forgetting in continual linear regression. Our experimental results show that an increase in forgetting is not unique to linear models and can be present when sequentially training with simple MLPs and even on more challenging continual learning problems.

This work raises a number of important questions for future study. First, the scenarios where replay provably increases forgetting are based on a few sequential tasks in a linear setting; to what extent can replay increase forgetting for a richer and more natural sequence of tasks? Secondly, can we provide a characterization of when replay increases forgetting? Moreover, can we give a necessary and/or sufficient conditions in terms of the quality/quantity of replayed samples that would mitigate forgetting? We show that the choice of the replayed samples matters, but these results do not provide a formal selection procedure, especially when the learner has no prior knowledge of future tasks.

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

# A ADDITIONAL BACKGROUND AND DEFINITIONS

## A.1 DERIVATION OF EQUATION 2

The following is based on Evron et al. (2022) and included here for completeness. More information on the properties of this solution and different forms of it are provided in their work.

We start with the closed form solution of Equation 1. Recall that $\mathbf{\Pi}_t, \boldsymbol{P}_t$ are orthogonal projections onto the spans of row spaces and null space of $\boldsymbol{X}_t$ respectively. Every solution to the equation $\boldsymbol{X}_t \boldsymbol{w} = \boldsymbol{y}_t$ can be written as $\boldsymbol{w} = \mathbf{\Pi}_t \boldsymbol{w}^* + \boldsymbol{P}_t \boldsymbol{v}$ for some vector $\boldsymbol{v}$. It is then easy to see that $\boldsymbol{P}_t w_{t-1}$ would minimize $\|\boldsymbol{w} - \boldsymbol{w}_{t-1}\|_2$ and is in the null space of $\boldsymbol{X}_t$, so the closed form solution would be $\boldsymbol{w}_t = \mathbf{\Pi}_t \boldsymbol{w}^* + \boldsymbol{P}_t \boldsymbol{w}_{t-1}$. Subtracting $\boldsymbol{w}^*$ from both sides would give Equation 2.

## A.2 DERIVATION OF EQUATION 4

This derivation is also in given in Evron et al. (2022) and is included here for completeness. Using Assumption 2.2, we can write each $\boldsymbol{y}_t = \boldsymbol{X}_t \boldsymbol{w}^*$ and the forgetting as

$$F_S(\boldsymbol{w}_T) = \frac{1}{T-1} \sum_{t=1}^{T-1} \|\boldsymbol{X}_t(\boldsymbol{w}_T - \boldsymbol{w}^*)\|_2^2. \tag{9}$$

Applying Equation 2 repeatedly, the parameter error vector after task $T$ would be

$$\boldsymbol{w}_T - \boldsymbol{w}^* = \boldsymbol{P}_T(\boldsymbol{w}_{T-1} - \boldsymbol{w}^*) = \boldsymbol{P}_T \ldots \boldsymbol{P}_1(\boldsymbol{w}_0 - \boldsymbol{w}^*) = -\boldsymbol{P}_T \ldots \boldsymbol{P}_1 \boldsymbol{w}^*. \tag{10}$$

Plugging the equation above into each term in forgetting, we have

$$F_S(\boldsymbol{w}_T) = \frac{1}{T-1} \sum_{t=1}^{T-1} \|\boldsymbol{X}_t(\boldsymbol{w}_T - \boldsymbol{w}^*)\|_2^2 = \frac{1}{T-1} \sum_{t=1}^{T-1} \|\boldsymbol{X}_t \boldsymbol{P}_T \boldsymbol{P}_{T-1} \ldots \boldsymbol{P}_1 \boldsymbol{w}^*\|_2^2. \tag{11}$$

## A.3 THE ANGLE BETWEEN NULL SPACES AND FORGETTING

Principal angles between two subspaces are a generalization of angles between two vectors. We start with the following simple two task example. Let $\boldsymbol{w}^*$ be an arbitrary unit vector and consider two tasks whose null spaces are $\boldsymbol{P}_1 = \boldsymbol{a}_1 \boldsymbol{a}_1^\top$ and $\boldsymbol{P}_2 = \boldsymbol{a}_2 \boldsymbol{a}_2^\top$, where $\boldsymbol{a}_1, \boldsymbol{a}_2$ are unit vectors such that $\|\boldsymbol{a}_1^\top \boldsymbol{w}^*\|_2 > 0$. In this two task case, the expected forgetting, given in Equation 7, is proportional to $\|\mathbf{\Pi}_1 \boldsymbol{P}_2 \boldsymbol{P}_1 \boldsymbol{w}^*\|_2^2$.

Note that $\boldsymbol{P}_1 \boldsymbol{w}^* = \boldsymbol{a}_1 \boldsymbol{a}_1^\top \boldsymbol{w}^* = c_1 \boldsymbol{a}_1$ and $\boldsymbol{P}_2 \boldsymbol{P}_1 \boldsymbol{w}^* = c_1 \boldsymbol{a}_2 \boldsymbol{a}_2^\top \boldsymbol{a}_1 = c_2 \boldsymbol{a}_2$ with $c_1 = \boldsymbol{a}_1^\top \boldsymbol{w}^*$, and $c_2 = c_1 \boldsymbol{a}_2^\top \boldsymbol{a}_1$. Figure 6 in Appendix A.3 shows how the norm of $\mathbf{\Pi}_1 \boldsymbol{P}_2 \boldsymbol{P}_1 \boldsymbol{w}^*$ depends on the angle between $\boldsymbol{a}_1$ and $\boldsymbol{a}_2$. When the angle between $\boldsymbol{a}_1$ and $\boldsymbol{a}_2$ is small, as in the left display, forgetting will be small. In fact if the angle was zero, then forgetting would be zero. At the other extreme is when the tasks are almost orthogonal, as shown in the right display, which would also result in less forgetting. Evron et al. (2022) show that the maximum forgetting (over the choice of $\boldsymbol{w}^*$ and $\boldsymbol{a}_2$) in this two task setting is proportional to $(\boldsymbol{a}_1^\top \boldsymbol{a}_2)^2 \big(1 - (\boldsymbol{a}_1^\top \boldsymbol{a}_2)^2\big)$ which is maximized when the angle between $\boldsymbol{a}_1$ and $\boldsymbol{a}_2$ is $\pi/4$.

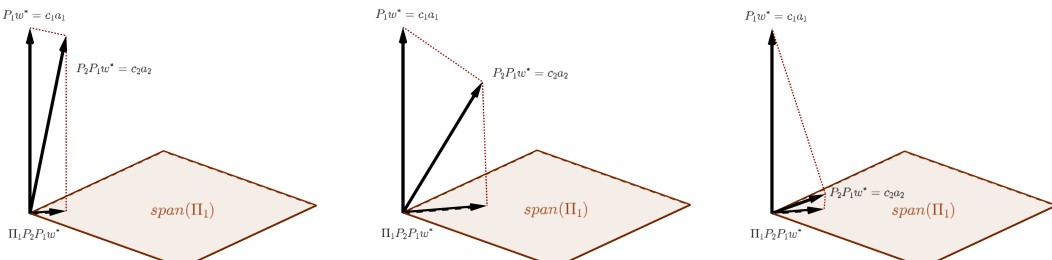

Figure 6: A simple example to demonstrate how the angle between tasks with one dimensional null spaces affects forgetting. $a_1, a_2$ span the null spaces of the first and second tasks respectively. The two vectors that are outside $\mathrm{span}(\Pi_1)$ show the projections $P_1 w^*$ and $P_2 P_1 w^*$ which will be in the directions of $a_1$ and $a_2$ as marked. Each display shows the effect of the angle between the null spaces, which is the angle between $a_1$ and $a_2$, on forgetting, which would be $\|\Pi_1 P_2 P_1 w^*\|_2$. As the angle between $a_1$ and $a_2$ increases (from the left to right display), the forgetting first increases and then decreases. Forgetting is maximized when the angle between $a_1$ and $a_2$ is $\pi/4$.

# B    Intuition for the proofs

## B.1    Intuition for the worst case result

Figure 7: Replay can transfer error between samples. The first task consists of two samples $x_1$ and $x_2$, while the second task has one sample $x_3$. All of the plots display the three samples $x_1, x_2, x_3$, the target parameter vector $w^*$, and the iterates without replay $w_1$ and $w_2$ from different angles. The vector $v_2$, which is orthogonal to the samples $x_1$ and $x_3$, is also displayed to show the orientation of the plots. The left figure in each row shows the general position of vectors of interest. Rest of the figures in the first row focus on the error of the final iterate $w_2$ on the first task's samples without replay. The second row, additionally, shows the final iterate $\tilde{w}_2$ after replay of $x_2$, and the projection of $\tilde{w}_2$ onto the first task's samples. In each case, intersection of the dashed line originating from parameter vectors $w^*, w_1, \dots$ with the sample $x_2$ or $x_1$ shows the projection of the parameter vector onto that sample. The error of each iterate $w_2$ and $\tilde{w}_2$ along a sample is marked in red by the discrepancy between its' projection and the projection of $w^*$ onto the sample.

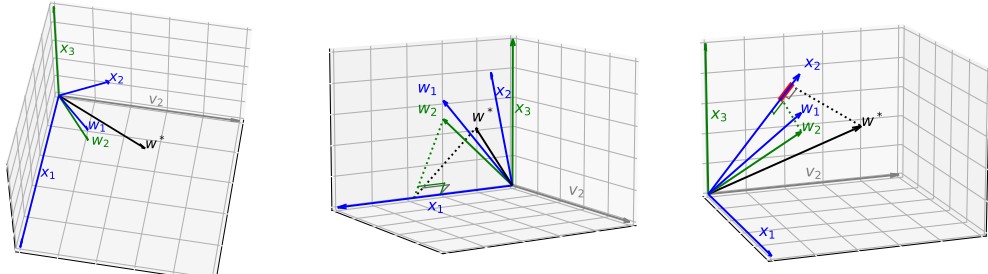

(a) Without replay: The first iterate $w_1$ is in the span of $x_1$ and $x_2$. The final iterate $w_2$ is obtained by training on $(x_3, x_3^\top w^*)$ starting from $w_1$. The change from $w_1$ to $w_2$ is in the direction of $-x_3$, and since $x_3$ is orthogonal to $x_1$, $w_2$ has no error along $x_1$. We can see this in the center figure where projections of $w_2$ and $w^*$ onto $x_1$ coincide. As we can see in the right figure, $w_2$ will have some error along $x_2$, since $x_3$ is not orthogonal to $x_2$. The discrepancy in the projections of $w_2$ and $w^*$ onto $x_2$ is shown in red.

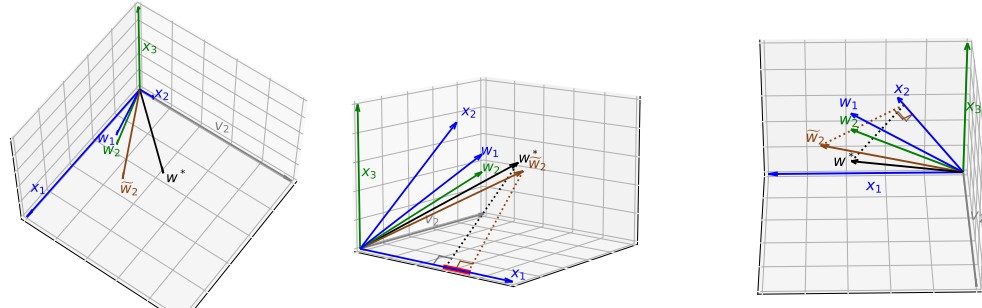

(b) With replay: Since $w_2$ had error on $x_2$, we consider replaying $x_2$. The first display shows the iterate after replay of $x_2$, $\tilde{w}_2$ relative to the other iterates and examples. Since $\tilde{w}_2$ changes in the direction of $x_2$ in addition to changing in the direction of $-x_3$, it also moves in the direction of $x_2$. This causes it to incur some error in the direction of $x_1$ as seen in the center figure. The right figure shows that in contrast to $w_2$, $\tilde{w}_2$ does not have any error along $x_2$, since $x_2$ was just replayed.

# C    Proofs

## C.1    Proofs of Main Results

**Proof of Theorem 3.2**    We construct a sequence of tasks where all the examples have unit norm. Fix an arbitrary orthonormal basis $v_1, \dots, v_d$ for $\mathbb{R}^d$ and consider the subspaces spanned by

$\{\boldsymbol{v}_1, \boldsymbol{v}_2, \boldsymbol{v}_3\}$ and $\mathbb{W} = \mathrm{span}\{\boldsymbol{v}_4, \ldots, \boldsymbol{v}_d\}$. Let

$$\boldsymbol{x}_1 = \boldsymbol{v}_1, \tag{12}$$

$$\boldsymbol{x}_2 = \frac{1}{2\sqrt{2}}\boldsymbol{v}_1 + \frac{1}{2\sqrt{2}}\boldsymbol{v}_2 + \frac{\sqrt{3}}{2}\boldsymbol{v}_3, \tag{13}$$

$$\boldsymbol{x}_3 = \boldsymbol{v}_3, \tag{14}$$

For each task $1 \le t \le T - 1$, let $\boldsymbol{X}'_t$ be a matrix containing a set of samples that are in the span of $\mathbb{W}$. That is, span of each $\boldsymbol{X}'_t$ is a subset of $\mathbb{W}$. The number of samples in each $\boldsymbol{X}'_t$ does not affect our construction.

For the first $T - 2$ tasks, set

$$\boldsymbol{X}_t = \begin{bmatrix} \boldsymbol{x}_1^\top \\ \boldsymbol{X}'_t \end{bmatrix}. \tag{15}$$

For the last two tasks we have

$$\boldsymbol{X}_{T-1} = \begin{bmatrix} \boldsymbol{x}_1^\top \\ \boldsymbol{x}_2^\top \\ \boldsymbol{X}'_{T-1} \end{bmatrix}, \tag{16}$$

and finally for the last task

$$\boldsymbol{X}_T = \begin{bmatrix} \boldsymbol{x}_3^\top \\ {\boldsymbol{w}'_1}^\top \\ \vdots \\ {\boldsymbol{w}'_{d-3}}^\top \end{bmatrix}, \tag{17}$$

where $\boldsymbol{w}'_1, \ldots \boldsymbol{w}'_{d-3}$ span $\mathbb{W}$. Let $\boldsymbol{u} = \sqrt{\frac{6}{7}}\boldsymbol{v}_2 - \frac{1}{\sqrt{7}}\boldsymbol{v}_3$ be a vector that is orthogonal to $\boldsymbol{x}_1$ and $\boldsymbol{x}_2$. We pick $\boldsymbol{w}^*$ such that $a := \boldsymbol{u}^\top \boldsymbol{w}^*$ is bounded away from zero. To compute forgetting without replay, we first compute $\boldsymbol{P}_T \boldsymbol{P}_{T-1} \ldots \boldsymbol{P}1\boldsymbol{w}^*$. For every $t < T - 1$, we decompose the null space $\boldsymbol{P}_t = \boldsymbol{P}'_t + \bar{\boldsymbol{P}}$, where $\bar{\boldsymbol{P}}$ is a projection into the span of $\boldsymbol{v}_2, \boldsymbol{v}_3$ (which are orthogonal to $\boldsymbol{x}_1$) and $\boldsymbol{P}'_t$ gives a projection of the null space $\boldsymbol{P}_t$ into $\mathbb{W}$. Note that $\boldsymbol{P}'_t$ and $\bar{\boldsymbol{P}}$ are projections into orthogonal subspaces, hence $\boldsymbol{P}'_t \bar{\boldsymbol{P}} = \bar{\boldsymbol{P}} \boldsymbol{P}'_t = 0$. For $t < T - 2$, we can write $\boldsymbol{P}_{t+1}\boldsymbol{P}_t = (\boldsymbol{P}'_{t+1} + \bar{\boldsymbol{P}})(\boldsymbol{P}'_t + \bar{\boldsymbol{P}}) = \boldsymbol{P}'_{t+1}\boldsymbol{P}'_t + \bar{\boldsymbol{P}}$. Then we have

$$\boldsymbol{P}_{T-2} \ldots \boldsymbol{P}_2 \boldsymbol{P}_1 \boldsymbol{w}^* = \boldsymbol{P}'_{T-2} \ldots \boldsymbol{P}'_2 \boldsymbol{P}'_1 \boldsymbol{w}^* + \bar{\boldsymbol{P}} \boldsymbol{w}^* \tag{18}$$

For the second to last task, we have $\boldsymbol{P}_{T-1} = \boldsymbol{u}\boldsymbol{u}^\top + \boldsymbol{P}'_{T-1}$, so $\boldsymbol{P}_{T-1} \ldots \boldsymbol{P}_1 \boldsymbol{w}^* = \boldsymbol{u}\boldsymbol{u}^\top \bar{\boldsymbol{P}} \boldsymbol{w}^* + \boldsymbol{P}'_{T-1} \ldots \boldsymbol{P}'_1 \boldsymbol{w}^*$. Since $\boldsymbol{w}'_1, \ldots \boldsymbol{w}'_{d-3}$ span $\mathbb{W}$, the last task's null space $\boldsymbol{P}_T$ is a projection into the subspace spanned by $\boldsymbol{v}_1, \boldsymbol{v}_2$, therefore, $\boldsymbol{P}_T \boldsymbol{P}'_{T-2} \ldots \boldsymbol{P}'_2 \boldsymbol{P}'_1 \boldsymbol{w}^* = 0$ and consequently

$$\boldsymbol{P}_T \boldsymbol{P}_{T-1} \ldots \boldsymbol{P}_1 \boldsymbol{w}^* = \boldsymbol{P}_T \boldsymbol{u}\boldsymbol{u}^\top \bar{\boldsymbol{P}} \boldsymbol{w}^*. \tag{19}$$

Since $\boldsymbol{u}$ is fully in the span of $\bar{\boldsymbol{P}}$, $\boldsymbol{u}\boldsymbol{u}^\top \bar{\boldsymbol{P}} = \boldsymbol{u}\boldsymbol{u}^\top$. Finally, the forgetting is

$$\frac{1}{T-1} \sum_{t=1}^{T-1} \left\| \boldsymbol{X}_t \boldsymbol{P}_T \boldsymbol{u}\boldsymbol{u}^\top \boldsymbol{w}^* \right\|_2^2. \tag{20}$$

We first compute forgetting on the samples $\boldsymbol{x}_1, \boldsymbol{x}_2$ and $\boldsymbol{x}_3$. Since $\boldsymbol{x}_3$ is included in the last task, $\boldsymbol{x}_3^\top \boldsymbol{P}_T \boldsymbol{u}\boldsymbol{u}^\top \boldsymbol{w}^* = 0$, and since $\boldsymbol{x}_1^\top \boldsymbol{P}_T = \boldsymbol{x}_1^\top$, we have $\boldsymbol{x}_1^\top \boldsymbol{P}_T \boldsymbol{u}\boldsymbol{u}^\top \boldsymbol{w}^* = \boldsymbol{x}_1^\top \boldsymbol{u}\boldsymbol{u}^\top \boldsymbol{w}^* = 0$. Now, for $\boldsymbol{x}_2$ we have

$$\boldsymbol{x}_2^\top \boldsymbol{P}_T \boldsymbol{u}\boldsymbol{u}^\top \boldsymbol{w}^* = \begin{bmatrix} \frac{1}{2\sqrt{2}} & \frac{1}{2\sqrt{2}} \end{bmatrix} \begin{bmatrix} \boldsymbol{v}_1^\top \\ \boldsymbol{v}_2^\top \end{bmatrix} \boldsymbol{u}\boldsymbol{u}^\top \boldsymbol{w}^* \tag{21}$$

$$= \frac{1}{2\sqrt{2}} \boldsymbol{v}_2^\top \boldsymbol{u}\boldsymbol{u}^\top \boldsymbol{w}^* = \frac{\sqrt{3}}{2\sqrt{7}} \cdot a. \tag{22}$$

The last equality follows from $v_2^\top u = \sqrt{\frac{6}{7}}$ and definition of $a$. The rest of the samples are in $\mathbb{W}$, and since the samples in $X_T$ span $\mathbb{W}$, we have that for all $i$, $X'_t P_T = 0$, that is, they make no contribution to forgetting. So the only sample that contributes to forgetting is $x_2$, which only occurs in task $T-1$. Then plugging in Equation 21 we have

$$\frac{1}{T-1}\sum_{t=1}^{T-1}\left\|X_t P_T u u^\top w^*\right\|_2^2 = \frac{1}{T-1}\left(x_2^\top P_T u u^\top w^*\right)^2 \tag{23}$$

$$= \frac{3}{28(T-1)}\cdot a^2, \tag{24}$$

which is of order $1/T$ as long as $a$ is bounded away from zero.

Now, we compare this to forgetting with replay of the sample $x_2, y_2$, which is the only sample that contributed to forgetting. In the replay scenario, $x_2$ is combined with $X_t$ to get $\tilde{X}_T$, consequently the null space $\tilde{P}_T = \frac{1}{2}(v_1 - v_2)(v_1 - v_2)^\top$. Recall that without replay $P_T = [v_1, v_2]\begin{bmatrix}v_1^\top \\ v_2^\top\end{bmatrix}$.

Now we compute the forgetting $\frac{1}{T-1}\sum_{t=1}^{T-1}\left\|X_t \tilde{P}_T u u^\top w^*\right\|_2^2$. Similar to the no replay case, we can see that for all $i$, $X'_t \tilde{P}_T = 0$ and $x_3 \tilde{P}_T = 0$, so these samples don't contribute to forgetting. Additionally, since $x_2$ has just been replayed, it is in the span of $\tilde{X}_T$ and doesn't contribute to forgetting. We are left with

$$x_1^\top \tilde{P}_T u u^\top w^* = \frac{1}{2}(v_1 - v_2)^\top u u^\top w^* = -\frac{6}{28}a \tag{25}$$

Forgetting with replay is then $\frac{T-1}{T-1}\left(x_1^\top \tilde{P}_T u u^\top w^*\right)^2 = \frac{9}{196}a^2$, since $x_1$ appeared in all the first $T-1$ tasks. Comparing $\frac{3a^2}{28(T-1)}$ to $\frac{9a^2}{196}$, we can see that as $T \to \infty$, forgetting vanishes without replay, while with replay, it is a constant. $\qquad\square$

**Proof of Theorem 3.6**   We use the following claim which simplifies the form of expected forgetting in the two task case with replay. Proof of this claim is given in Appendix C.2

**Claim C.1.** *Suppose that we have a sequence $S$ of two tasks in the average case setting (as described in Section 3.2). Let $R$ be a set of $m \le n_1$ randomly chosen (without replacement) indices of the samples to be replayed from the first task. Let $P_1, P_2$ be the task null spaces and $\tilde{P}_2 = \tilde{P}_2(\{x_{1,j}\}_{j\in R})$ be the null space of the second task with replay. Then the expected forgetting with respect to test samples and with replay of $m$ samples from the first task can be simplified to*

$$\mathbb{E}[F_{S'}(\tilde{w}_2)] = \mathbb{E}\left[\left\|\Pi_1 \tilde{P}_2 P_1 w^*\right\|_2^2\right], \tag{26}$$

*where the expectation is over the randomness of $\tilde{P}_2$.*

Given $w^*$, we can pick orthonormal basis $W_1$ and $W_2$ for the subspaces, $n_1$, and $n_2$. Also recall the sample generation process described in Equation 5. We start with describing the two task subspaces. We first fix $\epsilon = \sqrt{\frac{1}{63}}$, though other small values of $\epsilon$ would work as well. Then we fix an orthonormal basis $v_1, v_2, v_3$ such that $\|P_1 w^*\|_2 > 0$, where $P_1$ is defined in Equation 28. The first task is spanned by orthonormal vectors $v_1, u$, where $u = \epsilon v_2 + \sqrt{1 - \epsilon^2}v_3$ for some $1 > \epsilon > 0$ that will be fixed later. That is

$$\Pi_1 = W_1 W_1^\top = [v_1, u]\begin{bmatrix}v_1^\top \\ u^\top\end{bmatrix}. \tag{27}$$

This leads to the following one dimensional null space for the first task

$$P_1 = (\sqrt{1 - \epsilon^2}v_2 - \epsilon v_3)(\sqrt{1 - \epsilon^2}v_2 - \epsilon v_3)^\top. \tag{28}$$

The second task is spanned by $v_3$, leading to the null space

$$\boldsymbol{P}_2 = [\boldsymbol{v}_1, \boldsymbol{v}_2]\begin{bmatrix}\boldsymbol{v}_1^\top \\ \boldsymbol{v}_2^\top\end{bmatrix}. \tag{29}$$

Now we compute forgetting without replay using Equation 7. So we start with computing

$$\boldsymbol{\Pi}_1\boldsymbol{P}_2\boldsymbol{P}_1\boldsymbol{w}^* = \boldsymbol{v}_1\boldsymbol{v}_1^\top\boldsymbol{P}_2\boldsymbol{P}_1\boldsymbol{w}^* + \boldsymbol{u}\boldsymbol{u}^\top\boldsymbol{P}_2\boldsymbol{P}_1\boldsymbol{w}^* \tag{30}$$

$$= \boldsymbol{u}\boldsymbol{u}^\top\boldsymbol{P}_2\boldsymbol{P}_1\boldsymbol{w}^*. \tag{31}$$

The last equality holds since $\boldsymbol{v}_1^\top\boldsymbol{P}_2 = \boldsymbol{v}_1^\top$ and $\boldsymbol{v}_1^\top\boldsymbol{P}_1 = 0$. Let $a := ((\sqrt{1-\epsilon^2})\cdot\boldsymbol{v}_2 - \epsilon\cdot\boldsymbol{v}_3)^\top\boldsymbol{w}^*$. We have

$$\boldsymbol{u}\boldsymbol{u}^\top\boldsymbol{P}_2\boldsymbol{P}_1\boldsymbol{w}^* = \boldsymbol{u}[0, \epsilon]\begin{bmatrix}\boldsymbol{v}_1^\top \\ \boldsymbol{v}_2^\top\end{bmatrix}\boldsymbol{P}_1\boldsymbol{w}^* \tag{32}$$

$$= \epsilon\cdot a\cdot\boldsymbol{u}\boldsymbol{v}_2^\top(\sqrt{1-\epsilon^2}\boldsymbol{v}_2 - \epsilon\boldsymbol{v}_3) \tag{33}$$

$$= \epsilon\cdot\sqrt{1-\epsilon^2}\cdot a\cdot\boldsymbol{u}. \tag{34}$$

Forgetting without replay is

$$\|\boldsymbol{\Pi}_1\boldsymbol{P}_2\boldsymbol{P}_1\boldsymbol{w}^*\|_2^2 = \cdot\epsilon^2\cdot(1-\epsilon^2)\cdot a^2\cdot\|\boldsymbol{u}\|_2^2 \tag{35}$$

$$= \cdot\epsilon^2\cdot(1-\epsilon^2)\cdot a^2. \tag{36}$$

Note that forgetting on the last task is always zero. To compute forgetting with replay of one sample, we need to understand the distribution of $\tilde{P}_2$ first. Let $\mathbf{X}_{1J}$ be a randomly chosen sample from the first task. By Equation 5, $\mathbf{X}_{1J} = \boldsymbol{W}_1\mathbf{Z}_{1J}$ where $\mathbf{Z}_{1j} \sim \mathsf{N}(0, \frac{\mathbf{I}_2}{2})$ and $J \in_R [n_t]$ is an index that is picked uniformly at random from the set $[n_t]$. It's important here to note that $J$ and $\mathbf{Z}_{1J}$ are independent and $\mathbf{Z}_{1i}$ are iid. Then we have

$$p(\mathbf{Z}_{1J}) = \sum_{j=1}^{n_t} p(\mathbf{Z}_{1J} \mid J = j)p(J = j) \tag{37}$$

$$= \frac{1}{n_t}\sum_{j=1}^{n_t} p(\mathbf{Z}_{1J} \mid J = j) \tag{38}$$

$$= \frac{1}{n_t}\sum_{j=1}^{n_t} p(\mathbf{Z}_{1j}) = p(\mathbf{Z}_{11}), \tag{39}$$

and consequently $\mathbf{Z}_{1J} \sim \mathsf{N}(0, \frac{\mathbf{I}_2}{2})$ and we can write

$$\mathbf{X}_{1J} = \boldsymbol{W}_1\mathbf{Z}_{1J} = \frac{1}{2}(\alpha_1\boldsymbol{v}_1 + \alpha_2\boldsymbol{u}), \tag{40}$$

where $\frac{1}{2}\alpha_1, \frac{1}{2}\alpha_2 \sim \mathsf{N}(0, \frac{1}{2})$ are the two iid coordinates of $\mathbf{Z}_{1J}$.

$\tilde{P}_2$ is a projection onto the null space of the space spanned by task two samples, which have the form $\mathbf{X}_{2j} = \boldsymbol{v}_3\mathbf{Z}_{2j}$, plus $\mathbf{X}_{1J}$. In another words, since all the samples for task 2 are colinear with $\boldsymbol{v}_3$, $\tilde{P}_2$ is a projection onto the null space of linear span of $\{\boldsymbol{v}_3, \alpha_1\boldsymbol{v}_1 + \alpha_2\boldsymbol{u}\}$. Since $\alpha_1\boldsymbol{v}_1 + \alpha_2\boldsymbol{u} = \alpha_1\boldsymbol{v}_1 + \alpha_2\epsilon\boldsymbol{v}_2 + \alpha_2\sqrt{1-\epsilon^2}\boldsymbol{v}_3$, we have that

$$\mathrm{span}\left\{\boldsymbol{v}_3, \alpha_1\boldsymbol{v}_1 + \alpha_2\epsilon\boldsymbol{v}_2 + \alpha_2\sqrt{1-\epsilon^2}\boldsymbol{v}_3\right\} \tag{41}$$

$$= \mathrm{span}\{\boldsymbol{v}_3, \alpha_1\boldsymbol{v}_1 + \alpha_2\epsilon\boldsymbol{v}_2\} \tag{42}$$

Then $\tilde{P}_2$ is a projection onto a one dimensional vector space that is orthogonal to $\boldsymbol{v}_3$ and $\alpha_1\boldsymbol{v}_1 + \alpha_2\epsilon\boldsymbol{v}_2$. We can write

$$\tilde{\boldsymbol{P}}_2 = \frac{1}{z}\cdot(\alpha_2\epsilon\boldsymbol{v}_1 - \alpha_1\boldsymbol{v}_2)(\alpha_2\epsilon\boldsymbol{v}_1 - \alpha_1\boldsymbol{v}_2)^\top, \tag{43}$$

where $z := \epsilon^2 \alpha_2^2 + \alpha_1^2$ is a normalizing constant. Next we compute each of the terms in

$$\mathbf{\Pi}_1 \tilde{P}_2 P_1 w^* = v_1 v_1^\top \tilde{P}_2 P_1 w^* + u u^\top \tilde{P}_2 P_1 w^*. \tag{44}$$

We have

$$v_1 v_1^\top \tilde{P}_2 P_1 w^* = \frac{1}{z} \cdot v_1 v_1^\top (\alpha_2 \epsilon v_1 - \alpha_1 v_2)(\alpha_2 \epsilon v_1 - \alpha_1 v_2)^\top P_1 w^* \tag{45}$$

$$= \frac{\alpha_2 \epsilon}{z} \cdot v_1 (\alpha_2 \epsilon v_1 - \alpha_1 v_2)^\top P_1 w^* \tag{46}$$

$$= \frac{\alpha_2 \epsilon}{z} \cdot v_1 (\alpha_2 \epsilon v_1 - \alpha_1 v_2)^\top (\sqrt{1-\epsilon^2} v_2 - \epsilon v_3) \tag{47}$$

$$(\sqrt{1-\epsilon^2} v_2 - \epsilon v_3)^\top w^* \tag{48}$$

$$= -\frac{\alpha_1 \alpha_2 \epsilon \sqrt{1-\epsilon^2}}{z} \cdot a \cdot v_1, \tag{49}$$

and

$$u u^\top \tilde{P}_2 P_1 w^* = -\frac{\alpha_1 \epsilon}{z} \cdot u (\alpha_2 \epsilon v_1 - \alpha_1 v_2)^\top P_1 w^* \tag{50}$$

$$= \frac{\alpha_1^2 \epsilon \sqrt{1-\epsilon^2}}{z} \cdot a \cdot u. \tag{51}$$

Since $u$ and $v_1$ are orthogonal to each other, we can write

$$\left\| \mathbf{\Pi}_1 \tilde{P}_2 P_1 w^* \right\|_2^2 = \left\| u u^\top \tilde{P}_2 P_1 w^* \right\|_2^2 + \left\| v_1 v_1^\top \tilde{P}_2 P_1 w^* \right\|_2^2 \tag{52}$$

$$= \alpha_1^4 \epsilon^2 (1-\epsilon^2) \frac{a^2}{z^2} + \alpha_1^2 \alpha_2^2 \epsilon^2 (1-\epsilon^2) \frac{a^2}{z^2} \tag{53}$$

$$= \left( \frac{\alpha_1^2 \alpha_2^2 + \alpha_1^4}{z^2} \right) \epsilon^2 (1-\epsilon^2) a^2. \tag{54}$$

By Claim C.1, the expected forgetting with replay is

$$\mathbb{E}\left[ \left\| \mathbf{\Pi}_1 \tilde{P}_2 P_1 w^* \right\|_2^2 \right] = \mathbb{E}\left[ \frac{\alpha_1^2 \alpha_2^2 + \alpha_1^4}{z^2} \right] \epsilon^2 (1-\epsilon^2) a^2. \tag{55}$$

We compare this to expected forgetting without replay, which is $\epsilon^2 (1-\epsilon^2) a^2$, and show that there exists $\epsilon$ such that

$$\mathbb{E}\left[ \frac{\alpha_1^2 \alpha_2^2 + \alpha_1^4}{z^2} \right] > 1. \tag{56}$$

Note that by definition, $a^2 = \| P_1 w^* \|_2^2$ and the orthonormal basis $v_1, v_2, v_3$ can be chosen such that $a^2 > 0.$ .

By definition of $z$, can write $\frac{\alpha_2^2}{z} = (1 - \frac{\alpha_1^2}{z}) \cdot \frac{1}{\epsilon^2}$ and simplify the first term inside expectation to $\frac{\alpha_1^2 \alpha_2^2}{z^2} = \frac{\alpha_1^2}{z} \cdot \left(1 - \frac{\alpha_1^2}{z}\right) \cdot \frac{1}{\epsilon^2}$.

Let $\alpha_1'^2 = \frac{\alpha_1^2}{z}$, then rewriting Equation 56, we have picked $\epsilon$ such that by Claim C.2

$$\mathbb{E}\left[ \alpha_1'^2 \cdot (1 - \alpha_1'^2) \frac{1}{\epsilon^2} + \alpha_1'^4 \right] > 1. \tag{57}$$

We can then conclude that replay has increased average forgetting.

**Claim C.2.** *Let $\alpha_1, \alpha_2 \sim \mathsf{N}(0,1)$, and $\alpha_1'^{\,2} = \frac{\alpha_1^2}{\frac{\alpha_2^2}{63} + \alpha_1^2}$. Then*

$$\mathbb{E}\left[ 63 \alpha_1'^{\,2} - 62 \alpha_1'^{\,4} \right] \geq 1.4. \tag{58}$$

Proof of this claim is given in Appendix C.2.

$\square$

## C.2 PROOFS OF CLAIMS AND PROPOSITIONS

**Proof of Proposition 3.4**

Note that

$$\mathbb{E}\left[\mathbf{X'}_{tj}\mathbf{X'}_{tj}^{\top}\right] = \mathbb{E}\left[\boldsymbol{W}_t\mathbf{Z'}_{tj}\mathbf{Z'}_{tj}^{\top}\boldsymbol{W}_t^{\top}\right] = \boldsymbol{W}_t\frac{\mathbf{I}_k}{k_t}\boldsymbol{W}_t^{\top} = \frac{1}{k_t}\,\boldsymbol{W}_t\boldsymbol{W}_t^{\top} = \frac{1}{k_t}\boldsymbol{\Pi}_t, \tag{59}$$

where $\boldsymbol{\Pi}_i$ was the projection matrix into the subspace spanned by samples of task $i$. Additionally, we can write

$$\mathbb{E}\left[\mathbf{X'}_t^{\top}\mathbf{X'}_t\right] = \sum_{j\in[k_t]} \mathbb{E}\left[\mathbf{X'}_{tj}\mathbf{X'}_{tj}^{\top}\right] = \boldsymbol{\Pi}_t, \tag{60}$$

which will be useful when we compute expected forgetting below.

Recall that $\mathbf{Z}_{tj}$ were used to generate training samples for the task (Equation 5). Since any $k_t$ independent $\mathbf{Z}_{tj}$ samples are going to be linearly independent, we are guaranteed that $\mathbf{Z}_{tj}$ will have the same span as $\boldsymbol{W}_t$ under Assumption 3.3, which states that $n_t \geq k_t$. Consequently, the null space of each task $t$ is $\boldsymbol{P}_t = \mathbf{I} - \boldsymbol{\Pi}_t$. Then similar to Equation 4, we can write each term in the expected forgetting (with respect to test samples) as

$$\mathbb{E}[F_{S'}(\boldsymbol{w}_T)] = \frac{1}{T-1}\sum_{t=1}^{T-1}\mathbb{E}\left[\|\mathbf{X'}_t(\boldsymbol{w}_T - \boldsymbol{w}^*)\|_2^2\right] = \frac{1}{T-1}\sum_{t=1}^{T-1}\mathbb{E}\left[\|\mathbf{X'}_t\boldsymbol{P}_T\boldsymbol{P}_{T-1}\dots\boldsymbol{P}_1\boldsymbol{w}^*\|_2^2\right], \tag{61}$$

where now the expectation is only over the randomness in $X'_t$. Expanding the square inside the expectations in Equation 61 and applying Equation 60 we get

$$\mathbb{E}\left[\|\boldsymbol{X}_t\boldsymbol{P}_T\dots\boldsymbol{P}_1\boldsymbol{w}^*\|_2^2\right] = \mathbb{E}\left[\boldsymbol{w}^{*\top}\boldsymbol{P}_1\dots\boldsymbol{P}_T\boldsymbol{X}_t^{\top}\boldsymbol{X}_t\boldsymbol{P}_T\dots\boldsymbol{P}_1\boldsymbol{w}^*\right] = \boldsymbol{w}^{*\top}\boldsymbol{P}_1\dots\boldsymbol{P}_T\boldsymbol{\Pi}_t\boldsymbol{P}_T\dots\boldsymbol{P}_1\boldsymbol{w}^* \tag{62}$$

$$= \|\boldsymbol{\Pi}_t\boldsymbol{P}_T\dots\boldsymbol{P}_1\boldsymbol{w}^*\|_2^2, \tag{63}$$

where the last equality follows from the fact that $\boldsymbol{\Pi}_t$ is an orthonormal projection and $\boldsymbol{\Pi}_t = \boldsymbol{\Pi}_t^2$. Plugging this back into Equation 61 we can write the expected forgetting as

$$\mathbb{E}[F_{S'}(\boldsymbol{w}_T)] = \frac{1}{T-1}\sum_{t=1}^{T-1}\|\boldsymbol{\Pi}_t\boldsymbol{P}_T\dots\boldsymbol{P}_1\boldsymbol{w}^*\|_2^2. \tag{64}$$

$\square$

**Proof of Claim C.1**    Suppose that $m$ samples are randomly (without replacement) selected from the $n_1$ samples for task one. Alternatively, we can think of them as being fixed before seeing any samples. Let $S_1 \subseteq [n_1]$ be a randomly chosen set of indices of samples that were selected for replay. Let $\tilde{\boldsymbol{P}}_2$ be the projection into the null space of the combined samples for the second task. Then $\tilde{\boldsymbol{P}}_2 = \tilde{\boldsymbol{P}}_2(\{\mathbf{X}_{1,s}\}_{s\in S_1})$ is random, unlike $\boldsymbol{P}_2$. The expected forgetting is

$$\mathbb{E}[F_{S'}(\tilde{\boldsymbol{w}}_2)] = \mathbb{E}\left[\left\|\mathbf{X'}_1\tilde{\boldsymbol{P}}_2\boldsymbol{P}_1\boldsymbol{w}^*\right\|_2^2\right] = \mathbb{E}\left[\boldsymbol{w}^{*\top}\boldsymbol{P}_1\tilde{\boldsymbol{P}}_2\mathbf{X'}_1^{\top}\mathbf{X'}_1\tilde{\boldsymbol{P}}_2\boldsymbol{P}_1\boldsymbol{w}^*\right]. \tag{65}$$

Since $\{\mathbf{X}_{1,s}\}_{s \in S_1}$ is independent from $X_1'$ and $\mathbb{E}\left[\mathbf{X'}_1^\top \mathbf{X'}_1\right] = \mathbf{\Pi}_1$ (see Equation 60), we can write the expectation above as

$$\mathbb{E}\left[\boldsymbol{w}^{*\top} \boldsymbol{P}_1 \tilde{\boldsymbol{P}}_2 \mathbf{\Pi}_1 \tilde{\boldsymbol{P}}_2 \boldsymbol{P}_1 \boldsymbol{w}^*\right] = \mathbb{E}\left[\left\|\mathbf{\Pi}_1 \tilde{\boldsymbol{P}}_2 \boldsymbol{P}_1 \boldsymbol{w}^*\right\|_2^2\right], \tag{66}$$

where now the expectation is only over the randomness in $\tilde{\boldsymbol{P}}_2$.

$\square$

**Proof of Claim C.2**    Without loss of generality we can assume that $\alpha_1, \alpha_2 \sim \mathsf{N}(0,1)$, as this would not change the distribution of ${\alpha_1'}^2$. Define $f(\alpha_1') = 63{\alpha_1'}^2 - 62{\alpha_1'}^4 = {\alpha_1'}^2(63 - 62{\alpha_1'}^2)$. We can lower bound the expectation by considering the following three events:

1. ${\alpha_1'}^2 < 1/31$: under this event we use the trivial lower bound $f(\alpha_1') \geq 0$.

2. $1/31 \leq {\alpha_1'}^2 \leq 63/64$: then $f(\alpha_1') \geq 1.9$

3. ${\alpha_1'}^2 > 63/64$: since we always have ${\alpha_1'}^2 \leq 1$, we will use the bound $f(\alpha_1') \geq 1$.

Now we bound the probability of these events. Note that by symmetry, $\alpha_2^2 \leq \alpha_1^2$ with probability of $1/2$, then with would have

$$ {\alpha_1'}^2 = \frac{\alpha_1^2}{\frac{\alpha_2^2}{63} + \alpha_1^2} \geq \frac{\alpha_1^2}{\frac{\alpha_1^2}{63} + \alpha_1^2} = \frac{63}{64}. \tag{67}$$

So the event in Item 3 happens with probability $1/2$. Next, we argue that probability of the event in Item 1 is very small. If ${\alpha_1'}^2 = \frac{\alpha_1^2}{\frac{\alpha_2^2}{63} + \alpha_1^2} < 1/31$, then it must be that $30 \cdot 63\alpha_1^2 < \alpha_2^2$. We first argue that with high probability $\alpha_1^2 \geq \frac{4}{30 \cdot 61}$. Note that the pdf of normal distribution is upper bounded by $\frac{1}{\sqrt{2\pi}}$, so

$$\Pr\left[\alpha_1^2 < \frac{4}{30 \cdot 61}\right] \leq \sqrt{\frac{2 \cdot 4}{2\pi \cdot 30 \cdot 61}} \leq 0.018. \tag{68}$$

Then we have

$$\Pr\left[{\alpha_1'}^2 < 1/31\right] \leq \Pr\left[\alpha_1^2 < \frac{4}{30 \cdot 61}\right] + \Pr\left[30 \cdot 63\alpha_1^2 < \alpha_2^2 \mid \alpha_1^2 \geq \frac{4}{30 \cdot 61}\right] \tag{69}$$

$$\leq 0.018 + \Pr\left[30 \cdot 63\alpha_1^2 < \alpha_2^2 \mid \alpha_1^2 \geq \frac{4}{30 \cdot 61}\right]. \tag{70}$$

Note that

$$\Pr\left[30 \cdot 63\alpha_1^2 < \alpha_2^2 \mid \alpha_1^2 \geq \frac{4}{30 \cdot 61}\right] \leq \Pr\left[4 < \alpha_2^2\right] \leq 0.0001. \tag{71}$$

Now we have that $\Pr\left[{\alpha_1'}^2 < 1/31\right] \leq 0.019$. This lets us lower bound the probability of the event in Item 2 by $0.5 - 0.019 = 0.481$. Collecting these three bounds together we get

$$\mathbb{E}\left[63{\alpha_1'}^2 - 62{\alpha_1'}^4\right] \geq 0.481 \cdot 1.9 + 0.5 \cdot 1 \geq 1.4. \tag{72}$$

$\square$

# D    DETAILS OF THE EXPERIMENTS

## D.1    CONTINUAL LINEAR REGRESSION EXPERIMENTS

The linear models were trained starting from $0$ using SGD while the neural nets were trained with Adam and randomly initialized (Glorot uniform). The models are trained until convergence. Unless explicitly specified otherwise, the MLPs have one hidden layer of width $128d$ where $d$ is the input dimension. The number of samples per task was also $10$ and $100$ for the 3 and 50 dimensional case respectively.

**Training details**    Both of the linear and MLP models were trained for 7000 epochs on each task to produce Figure 2a, and 5000 epochs in Figure 2b. The batch sizes used for experiments with 3 and 50 dimensions are 4 and 32 respectively. The linear model in Figure 2a was trained with plain SGD with learning rate $0.1$. The linear model for the higher dimensional case in Figure 2b was also trained with SGD with learning rate 1 on the first task, while for the second task, the learning rate was $0.1$ with exponential decay rate $0.8$.

In the three dimensional case (Figure 2a), the MLP was trained on the first task using Adam with learning rate $8e{-}4$ and exponential decay rate $0.7$. On the second task, the learning rate was $0.001$ with exponential decay rate $0.6$. In the 50 dimensional case, the MLP was trained on the first task starting with learning rate $1e{-}4$ and exponential decay rate $0.6$. The starting learning rate on the second task was $0.001$ and the exponential decay rate was $0.6$.

These parameters were picked such that the training converges and training error is minimized. We have plotted the forgetting with higher number of independent runs, since the variance in error is quite high. Note that the statement of the average case result is on the expectation, and hence the error bars show standard mean error. The construction of the input distribution is the same as the one given in the proof of the theorem with $\epsilon = 0.2$.

**Replay Implementation**    Let $b$ be the batch size for training on the second task, during each training step of the second task, a random batch of $b' = \min\{b, m\}$ many samples from the second task are combined with a random batch of $b$ samples from $(\boldsymbol{X}^{\text{mem}}, \boldsymbol{y}^{\text{mem}})$, and they are weighted by $\frac{b}{b'}$ so that their total weight is equal to that of task two samples.

**Extension of the three dimensional construction in Theorem 3.6 to higher dimensions.**    Fix an arbitrary orthonormal basis $\{\boldsymbol{v}_1, \ldots, \boldsymbol{v}_d\}$ and $\epsilon = 0.4$. Set $\boldsymbol{u} = \epsilon \boldsymbol{v}_2 + \sqrt{1 - \epsilon^2}\boldsymbol{v}_3$ like the three dimensional construction. The first task spans the $d - 1$ dimensional subspace $\text{span}(\{\boldsymbol{v}_1, \boldsymbol{u}, \boldsymbol{v}_4, \ldots, \boldsymbol{v}_d\})$. As in the three dimensional construction, the second task is spanned by $\boldsymbol{v}_3$ only.

In both the three dimensional and higher dimensional case $d - 1$ samples from the first task would information theoretically be sufficient to learn $w^*$, but training until close to zero error might be challenging especially in the linear case. We experimentally verify this by directly solving the linear system and observing that replaying a few samples increases forgetting, while replaying 50 samples will result in zero forgetting.

**Narrower Network**    We also include Figure 8, which shows forgetting against the number of replay samples for a smaller network, where the width of the hidden layer is $4d = 200$. The input data is generated with the same distributional parameters as in Figure 2b. The training parameters for the second task were slightly different here. Specifically, the exponential learning decay rate used on the second task was $0.9$.

We note that it is possible that regularization, and training with small learning rate affect the observed pattern, especially with narrower networks. However, studying the effect of regularization and hyper parameters on forgetting with replay is outside the scope of this paper.

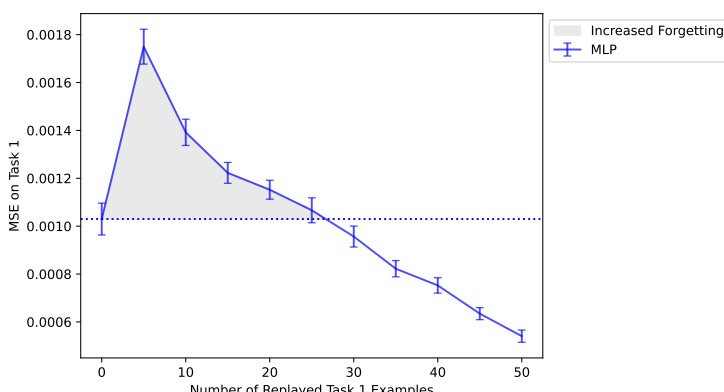

Figure 8: Same experiment as in Figure 2b with a network of width $4d$ instead

We have attached code used to generate input data for these experiments.

### D.1.1 THE EFFECT OF THE ANGLE BETWEEN TASKS WHILE TRAINING WITH A MLP

We discussed in Section 3.2 how replay changes the angle between the two tasks in a way that increases forgetting on average. To understand whether a similar mechanism is responsible for the increase in forgetting due to replay in the nonlinear case, we also look at the effect of the angle between two (linear) tasks on forgetting while using a nonlinear model for training. To do this, we pick the two tasks to be spanned by two $9$ dimensional subspaces in $\mathbb{R}^{10}$, so that their null spaces are essentially given by two vectors $\boldsymbol{u}_1, \boldsymbol{u}_2$. We vary the angle between $\boldsymbol{u}_1$ and $\boldsymbol{u}_2$ and for each angle measure forgetting on the first task after training on the second task, see Figure 9. We can see that forgetting of the MLP and linear model behave differently around angels that are close to $\pi/2$. In our three dimensional average case construction this won't matter, since initially without replay, the angle between the subspaces is close to zero, while with replay, the angle increases slightly but not a lot with high probability. Specifically, the construction is such that probability of replay leading to the angle being close to $\pi/2$ is very small.

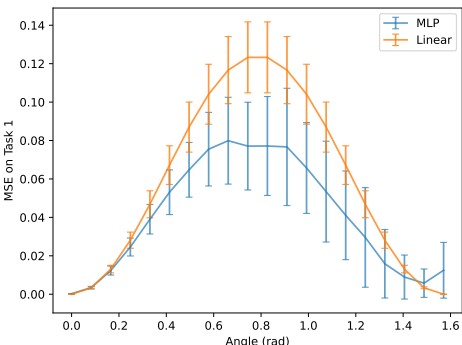

Figure 9: When training using a MLP, the angle between task null spaces mostly has a similar effect on forgetting as the linear case, with the exception of near orthogonal angles. Each point is averaged over 50 runs and the error bars here show standard deviation.

Here we give more details on the experiments used to get Figure 9. The input distributions and what we referred to as the angle between two tasks has been already discussed in the last paragraph of Section 4. The number of samples used per task is $100$. The linear model was trained using SGD with learning rate $0.1$. During training on the second task there was exponential decay rate of $0.95$. The MLP had one hidden layer of width $128d$, and it was trained on the first task with starting learning

rate $1e - 4$ and on the second task with initial learning rate $0.001$. in both cases (MLP training on task one and two), there was an exponential decay rate $0.6$. All the models for this experiment were trained for 5000 epochs with the batch size 32.

## D.2 EXPERIMENTS ON MNIST

In all the experiments, a fully connected network with two hidden layers of size 256 was used. In all cases, training on each task was for 3 epochs, with batch size of 32, and using Adam (Kingma & Ba, 2014) with learning rate of $0.001$.

**Statistical tests.** When we compared the means, we used Welch's t-test, which is similar to a student t-test while allowing the populations to have different variances.

**Rotated MNIST.** Rotated MNIST experiments are in a task incremental setting and use the training data for all the 10 digits. The training data on the second task is the same as the training data on the first task, except that it is rotated. Forgetting is measure on test samples. The replay is done the same as the regression experiments. That is, for each class, a random sample is combined with the samples in the each batch and the replay samples are up-weighted such that the replay sample has equal weight to the rest of the samples. The no replay baseline is what the literature might call the fine tuning baseline. The network is sequentially trained on the two tasks. The optimizer is reset after training on the first task.

**Split MNIST.** These experiments are in a class incremental setting, so the network had 4 output heads. During evaluation on the first task, only the logits corresponding to the classes in the first task were taken into account. This is the case with or without replay. Again, the replay implementation here is similar to the regression experiments.

## D.3 COMPUTE RESOURCES

### D.3.1 REGRESSION EXPERIMENTS

The experiment in Figure 9 took about 20 hours on a machine with single NVIDIA GeForce RTX 4080 GPU. Each run of the experiments in figures 2 and 8 would take about $0.5 - 1$ hour on a single NVIDIA A100-SXM4-80GB GPU. All the experiments did not use a significant amount of memory, since the input data was at most 50 dimensional.

## D.4 MNIST EXPERIMENTS

The experiments in Figure 3 took about 6 hours on a machine with a single NVIDIA GeForce RTX 4080 GPU for each rotation. The experiment in Figure 5 took about 2 hours on the same machine.

