# OpenReview forum: "Replay can provably increase forgetting"
_ICLR.cc/2025/Conference — ICLR 2025 Conference Withdrawn Submission_

### Official Review · Reviewer_f4cg · 2024-10-26

**Soundness:** 3
**Presentation:** 3
**Contribution:** 2
**Rating:** 3
**Confidence:** 4

**Summary:**

This paper examines the performance of replay-based methods in an over-parameterized linear continual learning (CL) model through both theoretical analysis and experiments. The findings reveal an uncommon phenomenon: replay may negatively impact CL performance. The theoretical analysis illustrates this phenomenon in two scenarios—adversarial and random replay. In experiments, the theoretical results are verified and extended to real datasets, demonstrating that the negative effects of replay can also be observed in practice.

**Strengths:**

They studied the negative impact of replay in CL, which is indeed a surprising topic. Their theoretical results explained the reason of the negative impact, which is further verified in experiments.

**Weaknesses:**

Both theoretical(Theorem 3.6) and experimental results focus on T=2, which is not general enough. Theorem 3.2 is an extreme example, which makes the result less surprising. Furthermore, the experimental parts should at least present what will happen when T is large than 2.

**Questions:**

After reading your paper, I have the following questions and suggestions.
1. As I'm concerned in weaknesses, Theorem 3.2 appears to be rather extreme(two taks with three samples), making it unsurprising that there exist certain cases where replay may negatively impact the performance related to forgetting. Can you provide addtional analysis to illustrate this observation under a more general problem setup?

2. Theorem 3.6 focus the average performance in the case of only two tasks. Similar to Theorem 3.2, it demonstrates only the existence of an example, without providing a more detailed illustration of when this phenomenon occurs. As far as I know, [1] has present a closed form of forgetting and generalization error for T=2 in a similar problem setup. Based on this, one can observe that the performance of forgetting is not monotonic with respect to the size of replay. This can be interpreted as a more comprehensive conclusion regarding the impact of replay. In comparison to the aforementioned conclusion from their work, could you provide additional insights derived from your own conclusions?

3. The experiments focus on the case when T=2, which may not provide sufficient insight for practical applications. Could you conduct additional experiments for scenarios where T>2?

4. A key question that significantly influences my assessment of your paper is: Based on your theoretical and empirical findings, can you provide a more systematic characterization or discussion of the conditions under which replay is detrimental versus beneficial?

Reference:
[1] Banayeeanzade, M., Soltanolkotabi, M., & Rostami, M. (2024). Theoretical Insights into Overparameterized Models in Multi-Task and Replay-Based Continual Learning. arXiv preprint arXiv:2408.16939.

---

> ### Author Response · Authors · 2024-11-25
>
> We thank the reviewer for their time and feedback.
> Please see the general comments for the main weakness.
>
> **Questions**
> 1. It seems that there is a misunderstanding about the notation. Theorem 3.2 is for any number of tasks (T) greater than two  and dimensionality (d) of at least 3.
> 2. We will add a paragraph to compare our results to those in [1]. The setting in [1] is linear, but still quite different from ours. The most significant difference is that they assume the samples for all tasks are from a normal distribution, which means that the subspaces for each task are random. This allows them to get more general expressions, while the setting would not take into account the role of the relationship between task subspaces. The setup in [1] also allows the tasks to have different linear predictors, so there is no joint realizability assumption. Even though this is a strictly weaker assumption, it would introduce error that is not due to the sequential nature of the problem but rather the fact that there might not exist a linear predictor that could do well on all tasks.
>
>     Regarding the monotonicity of forgetting with replay, it seems that the reviewer is referring to equation 17 in Theorem 4.2 of [1]. Under the joint realizability assumption that we have, using their notation, $w^*_1$ = $w^*_2$, which means that the first term in equation 17 would be zero, and we would be left with the monotonically decreasing term. Indeed, part of the reason that our results are surprising is that we see this even though the tasks share the same linear predictor.
>
>  3. We could conduct experiments with more tasks, however, as a starting point, it is not unusual to just consider two tasks for simplicity, for example, see [2].
>
>  4. Unfortunately, a full characterization of conditions under which replay would increase forgetting seems challenging in this setting. We do hint at one such condition in Remark 3.8.
>
> **References**
>
> [2] Daniel Goldfarb and Paul Hand. Analysis of catastrophic forgetting for random orthogonal transformation tasks in the overparameterized regime. AISTATS 2023

---

### Official Review · Reviewer_aJJx · 2024-10-29

**Soundness:** 2
**Presentation:** 3
**Contribution:** 2
**Rating:** 3
**Confidence:** 4

**Summary:**

The best-performing methods for the Continual Learning problem are currently memory-based. These methods save a percentage of examples from previous tasks that can be used in subsequent tasks. The authors perform a theoretical analysis of how example selection affects forgetting. The authors show that, given certain assumptions and constraints, there are scenarios where specific examples can cause more forgetting than not having them in the buffer. The authors show that this is particularly critical when the buffer is limited to a small number of samples per task. The authors also provide an empirical analysis using MNIST.

**Strengths:**

- Theoretical analyses in the area of CL are scarce. The authors identified a gap between previous work and current methods, which can help to understand the limitations and strengths of current memory-based methods, as well as help to understand some empirical results that may be unintuitive.
- The work has a clear motivation, and the authors identified a need to increase the theoretical understanding in this research area.

**Weaknesses:**

- I agree with some of the authors' conclusions, but they ignore an essential body of work in CL that focused on the selection of items stored in memory. Although many of these papers focus on empirical studies, they reach similar and, in some cases, more robust conclusions than those found in this paper.
    - The authors' analysis is based on simple models and scenarios, which can often be difficult to extrapolate to more complex scenarios. If empirical results are presented, I recommend also presenting more complex scenarios (e.g., TinyImageNet) to make the results more robust. This is particularly important since it is well-known that different models and methods behave differently in more complex scenarios.
    - This is even more crucial when the assumptions of the scenarios presented in the analyses are rarely presented in more complex situations, which reduces the impact of the analyses presented.
- Something that I cannot entirely agree with the authors, but I think maybe a problem of definitions, is presented in lines 170-172, which is then stated in Remark 3.7. I do believe that it can be a problem of overfitting since in high dimensionality problems, when replaying a model with a single example, the model will tend to over-represent this sub-space, and if the class or task is representable by multiple spaces within the distribution (which is a common issue), these will be forgotten due to the overfitting of this example to a particular sub-space. In this case, overfitting and interference may be related,  but they do not rule it out completely, as the authors do. Could the authors clarify the definition of interference or overfitting?
    - The problem lies in how the whole work is presented since the authors speak of sub-spaces that generate forgetting, which can be seen as an overfitting of the selected example. It would be good if the authors could analyse why they believe it is not overfitting.

**Questions:**

- From the results presented. How feasible is it that these can be extrapolated to:
    - More complex scenarios where the orthogonality assumption is not met?
    - Scenarios where the distribution of tasks/classes is not uniform? Or is it more complex, for example, when each task/class may consist of multiple sub-spaces?
    - For example, in worst-case analysis, can the adversarial example be replaced by one outside or away from the training distribution? This commonly happens in complex datasets. How similar would these results or analyses be to those presented in the paper?
- How are the task data generated in the analysis?
    - Suppose they are created from simple distributions (such as a Gaussian) without a possible relationship between the classes. In that case, it is easy to see that the interference between the tasks can be low. The restriction of the assumptions may affect the analysis presented.
    - How does the analysis behave with a possible correlation of the distributions of the different tasks?
- Can your analysis be used to propose a new method to populate the memory?

---

> ### Author Response · Authors · 2024-11-25
>
> We thank the reviewer for their time and feedback.
>
> - The goal of this paper was to gain a theoretical understanding of replay.
>     As a first step towards understanding more complex scenarios, it is useful to analyze simpler settings first.
>     It is common to use linear models to study neural networks in theoretical works due to tractability issues. This is further justified in the NTK regime. For justification of other assumptions in this work, please see the points under item 1 of response to reviewer Z4ns.
>
> -  Thanks for pointing this out. We will add clarification to this remark. The point we want to highlight here is that there is no label noise in our setup.
>     - Typically, overfitting refers to a model performing well on the training set by fitting to noise while doing poorly on the test set. There is no noise in our setting.
>     - In terms of over-representing certain sections of the subspace, consider the following scenario. After training on the first task is done, instead of training on samples from the second task plus replay samples, train only using the replay samples. This would not introduce any error on the first task, which would conflict with the intuition that the replay samples are over represented.
>     Another way to see this is to note that the model has the capacity to fit all samples perfectly, so it does not have to be forced to balance performance on different samples.
>
>
> ** Questions **
> - More complex scenarios:
>     -  We don't have an orthogonality assumption. Could the reviewer clarify which assumption they are referring to?
>     - In our setup we don't assume a distribution over tasks or classes. Since
>         we are giving negative results, our constructions would still be a valid instance of these more complex setups.
>
>     - It would be possible to replace the adversarial example with an example that has not been seen so far. In fact that would make it easier to form these constructions. However, it would be hard to justify that as replay.
>
> - (Task data generation)
>     - Data is sampled from Gaussians supported in the given subspaces. Assuming that the data is Gaussian (supported in the full space) with fixed sample size would imply that essentially each subspace is picked uniformly at random. Depending on dimensionality of the ambient space it could mean that there would be less interference. However, this setting would be restrictive since it would would not take into account how different interactions between subspaces could affect forgetting. In this setting, we are essentially not making any assumptions on how the task subspaces are related, and asking: Can they be related such that replay would increase forgetting?
>
>     -  What does it mean for distributions of different tasks to be correlated? If this is referring to the orthogonality comment, there is a misunderstanding here. The task subspaces we have in the contructions are not orthogonal to each other.
>
> - We do not if/how this analysis can be used for selection of samples, especially without advanced knowledge of future tasks.

---

> > ### Comment · Reviewer_aJJx · 2024-11-27
> >
> > I appreciate the authors' response.
> >
> > Given the feedback from the other reviewers and the authors' response, I agree that there is a big difference between what each reviewer understood and what the authors meant to convey, demonstrating a bit of a deficiency in the work in terms of how they wrote what they wanted to convey.
> >
> > I agree with the authors that the paper shows a possible limitation in memory-based methods. However, I agree with reviewer Z4ns that their assumptions are too strong, which limits their results. This can be critical in areas where the main motivation is to apply the methods in real environments, such as CL.
> >
> > To clarify the orthogonality issue, I did not refer to MNIST's assumption of orthogonality but to the worst-case scenario demonstrations in Section 3.
> > Having said that, the authors show that the negative effect can happen in MNIST. However, I think it can also be shown that it can occur in more complex scenarios (like Tiny Imagent)if one uses adversarial examples to populate the memory. This could strengthen the empirical results and give validity to the theoretical ones.
> > To clarify the point of using unseen examples, I meant examples that may be noisy, difficult, or slow to learn. Other work has shown that adding these examples to memory can affect the performance of memory-based methods. All these results in benchmarks with more complex distributions.
> >
> > With the above, I link to methods that study how to populate memory. Some of these papers study the best (or worst) examples to store based on different criteria. The work presented here can be expanded along these lines, adding further empirical validity and possibly providing a further theoretical contribution.

---

### Official Review · Reviewer_qSWt · 2024-10-30

**Soundness:** 2
**Presentation:** 4
**Contribution:** 1
**Rating:** 3
**Confidence:** 4

**Summary:**

The paper analyzes replay in the realizable linear regression setting (Evron et al. 2022). The paper shows formally that replay can increase forgetting in two settings: a worst-case setting for single samples, and an average case for appropriate task sequences. Toy experiments match the theoretical results, showing increased forgetting for low replay sizes (up to 3 samples in total).

**Strengths:**

- The paper is well written and easy to read.
- The paper tackles an important theoretical question, that is the problem of how to analyze replay-based continual learning.

**Weaknesses:**

The paper makes a very strong statement: "Replay can provably increase forgetting". Results on replay may be the most robust piece of evidence in the whole continual learning literature. Therefore, I expect that claiming that replay provably increases forgetting requires exceptional evidence. I would argue that the results of the paper are more a property of the extremely limited setting than a general property of replay-based methods.
- (line 65) the paper argues that it is counter-intuitive that there are worst-case settings. However, the paper only considers very low memory sizes. It is well known that intra-task interference is an issue, as shown in the paper. Therefore, the results are not counter-intuitive but agree with the literature.
- (theoretical analysis) It is interesting to see that the analysis still somewhat generalizes to the "average-case" setting. However, this is still a very artificial setting. No one expects replay to work with a single example. There are empirical results showing it may be beneficial, but those only demonstrate that very small replay buffers may work, not that they should.
- (experimental evidence) The experiment on MNIST uses replay on a single class or on very few samples (forgetting only happens with less than 3 samples). Again, forgetting in this setting is not exactly surprising and it is not sufficient evidence for the main claim.

The paper claims that "forgetting is a property of the task rather than the model", which I don't think is sufficiently supported by the evidence, given that it only happens in a very limited setting (Split MNIST with very limited replay). Of course forgetting is partially a consequence of task interference (or intra-task interference in this case), but data does not fully explain forgetting.

Overall, I think the paper should be generalized to stronger results in a more general setting to be a useful theoretical analysis of replay.

minor:
- I think assumption labels are not updated
- L115 typo: Inverse of X_t

**Questions:**

See weaknesses above.

---

> ### Author Response · Authors · 2024-11-25
>
> We thank the reviewer for their time and feedback.
> Regarding the main weakness, it seems that there has been a misunderstanding about what the main message of this paper is. Please see the general comment.
>
> - (line 65) It is true that we mostly see this with smaller replay sample sizes, though for one experiment (Figure 2 (b)), it is also observed for larger sample sizes.
>     It would be very helpful if the reviewer could provide some references to the literature that makes these results not counter-intuitive. It's important to note that in our regression setting, the samples are not noisy and all are realized by the same linear function.
> - (Theoretical analysis) This goes back to the point addressed in the general comment.
>     Additionally, as far as we know, the existing literature does not consider the specific outcome of replay of few examples hurting performance. But, again, this is not the main point of this paper.
>
> -  (Experimental evidence) The experimental evidence also include the regression experiments, including Figure 2(b), where a larger number of replay samples increases forgetting. Please note that as described in the general comment, the main claim of the paper is not that replay of a few samples will increase forgetting. Also the interesting part of Figure 5 (which the reviewer is referring to) is that for one of the task sequences, replay  increases forgetting, while for the other one it doesn't. Note that these task sequences share the same classes.
>
>
> We understand that this sentence in the abstract made too strong of a claim, we meant to highlight that the effects of replay that we observed in the linear setting can be robust to some extent to the choice of model used for learning. This was essentially referring to the experiments in Figure 2, where a MLP behaves quite similarly to the linear model on the same inputs when it comes to forgetting. We will clarify this part of the abstract. Thanks for pointing it out.

---

### Official Review · Reviewer_Z4ns · 2024-11-03

**Soundness:** 3
**Presentation:** 3
**Contribution:** 2
**Rating:** 3
**Confidence:** 4

**Summary:**

This paper provides a theoretical study on replay-based continual learning (CL) on overparameterized linear models. In particular, this paper shows that replaying with a single example can indeed be harmful, i.e., increasing the forgetting, under certain scenarios, by constructing task sequences appropriately. Theoretical results are provided for multiple cases. Moreover, experiments are conducted on both linear models and MLPs to verify the theoretical insights.

**Strengths:**

1. Theory on replay-based CL is very limited and this paper provides an attempt in this direction to understand when replaying one examplar can increase forgetting.

2. The theoretical results are verified in the experiments.

3. The presentation is clear.

**Weaknesses:**

1. The assumptions are too strong, particularly assumption 2.2. It is hard to justify how this can be true in practice. The assumptions 3.1 and 3.3 are also restrictive. For overparameterized linear models, assuming a constant sample norm is not widely seen. Given the fact that investigating linear models is already a restrictive setup, making these additional assumptions further weakens the importance of the theoretical results.

2. The definition of forgetting in equation 3 is based the training samples, which does not make sense at all.

3. While the average case uses a more standard definition of forgetting on test samples, the theoretical results is only limited to a simple two-task setup.

4. The experiments are only very limited, especially for the setup with neural networks. While the theory might take more efforts with more than 2 tasks, the experiments can be easily done with more tasks, which is not the case in the paper. Besides, to convince readers that the theoretical insights can also be observed in neural networks, more experiments on larger dataset such as CIFAR should be conducted. While the theoretical results have been observed in certain scenarios, the small-scale experimental verification cannot sufficiently convince the readers that replaying with a small number of samples is often harmful in CL. Even in the experiments in this paper, replaying with one sample usually helps.

5. It is not clear how useful the theoretical results in this paper can be, as they highly depend on deliberate constructions of tasks and data.

**Questions:**

Please see the weaknesses above. One more question is how you handle the correlation between learned model and replay data in the average case in theory.

---

> ### Author Response · Authors · 2024-11-25
>
> We thank the reviewer for their input.
> We understand that the reviewer finds the assumptions too strong.
> In the next version we will include
> more context/justification for these assumptions in the paper. We start with addressing that:
> 1. Background/justification of assumptions.
>     * Assumption 2.2 (Joint Realizability), is common in the related literature [1], [2], [3] and serves the following two purposes. First, from a practical point of view, it is close to scenarios where multitask learning with a large network could lead to close to zero training error (for example, MNIST, and over-parametrized networks in general). Second, it allows one to isolate the problem of forgetting because of learning continually, rather than the linear model not having the capacity to learn all the tasks, which could happen if each task was realized by different parameters. Under this assumption joint training on all tasks would lead to zero error.
>
>     * Assumption 3.1 (constant sample norm) is used in Theorem 3.2 but is not necessary to get the results in Theorem 3.2. We included it to emphasize that the norm of the samples does not matter. We could add a remark about this or adjust the construction so that this assumption is not needed.
>
>     * Assumption 3.3 can be thought of as saying that the data distribution is not full rank. In general, when the number of samples is less than the dimension, they will span some subspace, depending on the distribution.
>
> 2. Could the reviewer elaborate more on why this doesn't make sense?
>      Usually test error is larger than training error and this definition of  forgetting based on training samples makes it more surprising that the model's performance degrades on samples it has already trained on.  Additionally, this definition has been used in existing literature [1].
>
> 3. Two tasks: To present negative results that highlight conceptual limitations of replay, the focus has been to provide minimal constructions that would prove the point. There is also related work where the results involve only two tasks [3].
>
> 4. and 5 : Please see the general comment.
>
>
>
>
> **Questions**
> The learned model does depend on the replay sample, which is random in the average case setup. We find the distribution of the learned parameters and use that to bound expected forgetting. Please see proof of Theorem 3.6 for details.
>
> **References**
>
> [1] Itay Evron, Edward Moroshko, Rachel Ward, Nathan Srebro, and Daniel Soudry. How catastrophic can catastrophic forgetting be in linear regression? COLT 2022.
>
> [2] Itay Evron, Edward Moroshko, Gon Buzaglo, Maroun Khriesh, Badea Marjieh, Nathan Srebro, and Daniel Soudry. Continual learning in linear classification on separable data. ICML 2023.
>
> [3] Daniel Goldfarb and Paul Hand. Analysis of catastrophic forgetting for random orthogonal transforma- tion tasks in the overparameterized regime. AISTATS 2023

---

### Author Response · Authors · 2024-11-25

We thank the reviewers for their time and feedback. The reviewers highlighted the following strengths of the paper.
- Addresses a current gap in literature in theoretical understanding of replay.
- Theoretical results are verified empirically.
-  It is well-written and the presentation is clear.

Given the setting we are in, where all the tasks share the same parameters and there is no label noise, we found the results counter-intuitive.
Here we address a point that was shared across some of the reviews, and seems to have affected further evaluation of the paper.
The raised point was that the paper does not provide enough evidence that replay will increase forgetting.
The main claim of the paper is not that replay *will* increase forgetting, but rather that it could. The aim of this paper has been to understand sample replay theoretically, and this includes its' limitations, which we found surprising. It is not uncommon for negative results to be based on a construction since they usually have an existential nature. We have tried to probe to some extent to see how far these limitations can be realized in more practically relevant settings, but we are not claiming that these limitations are frequently observed in practice.

---

### Note · Authors · 2024-12-04

**Comment:**

We thank the reviewers for sharing their perspectives and will take their comments into account in a future revision of the paper.

**Withdrawal Confirmation:**

I have read and agree with the venue's withdrawal policy on behalf of myself and my co-authors.